# Precise expressions for random projections: Low-rank approximation and randomized Newton

**Michał Dereziński**
Department of Statistics
University of California, Berkeley
mderezin@berkeley.edu

**Feynman Liang**
Department of Statistics
University of California, Berkeley
feynman@berkeley.edu

**Zhenyu Liao**
ICSI and Department of Statistics
University of California, Berkeley
zhenyu.liao@berkeley.edu

**Michael W. Mahoney**
ICSI and Department of Statistics
University of California, Berkeley
mmahoney@stat.berkeley.edu

## Abstract

It is often desirable to reduce the dimensionality of a large dataset by projecting it onto a low-dimensional subspace. Matrix sketching has emerged as a powerful technique for performing such dimensionality reduction very efficiently. Even though there is an extensive literature on the worst-case performance of sketching, existing guarantees are typically very different from what is observed in practice. We exploit recent developments in the spectral analysis of random matrices to develop novel techniques that provide provably accurate expressions for the expected value of random projection matrices obtained via sketching. These expressions can be used to characterize the performance of dimensionality reduction in a variety of common machine learning tasks, ranging from low-rank approximation to iterative stochastic optimization. Our results apply to several popular sketching methods, including Gaussian and Rademacher sketches, and they enable precise analysis of these methods in terms of spectral properties of the data. Empirical results show that the expressions we derive reflect the practical performance of these sketching methods, down to lower-order effects and even constant factors.

## 1 Introduction

Many settings in modern machine learning, optimization and scientific computing require us to work with data matrices that are so large that some form of dimensionality reduction is a necessary component of the process. One of the most popular families of methods for dimensionality reduction, coming from the literature on Randomized Numerical Linear Algebra (RandNLA), consists of data-oblivious sketches [Mah11, HMT11, Woo14]. Consider a large $m \times n$ matrix $\mathbf{A}$. A *data-oblivious sketch* of size $k$ is the matrix $\mathbf{SA}$, where $\mathbf{S}$ is a $k \times m$ random matrix such that $\mathbb{E}[\frac{1}{k}\mathbf{S}^\top\mathbf{S}] = \mathbf{I}$, whose distribution does not depend on $\mathbf{A}$. This sketch reduces the first dimension of $\mathbf{A}$ from $m$ to a much smaller $k$ (we assume without loss of generality that $k \ll n \leq m$), and an analogous procedure can be defined for reducing the second dimension as well. This approximate representation of $\mathbf{A}$ is central to many algorithms in areas such as linear regression, low-rank approximation, kernel methods, and iterative second-order optimization. While there is a long line of research aimed at bounding the worst-case approximation error of such representations, these bounds are often too loose to reflect accurately the practical performance of these methods. In this paper, we develop new theory which enables more precise analysis of the accuracy of sketched data representations.

A common way to measure the accuracy of the sketch $\mathbf{SA}$ is by considering the $k$-dimensional subspace spanned by its rows. The goal of the sketch is to choose a subspace that best aligns with the distribution of all of the $m$ rows of $\mathbf{A}$ in $\mathbb{R}^n$. Intuitively, our goal is to minimize the (norm of the) residual when projecting a vector $\mathbf{a} \in \mathbb{R}^n$ onto that subspace, i.e., $\mathbf{a} - \mathbf{Pa} = (\mathbf{I} - \mathbf{P})\mathbf{a}$, where $\mathbf{P} = (\mathbf{SA})^\dagger \mathbf{SA}$ is the orthogonal projection matrix onto the subspace spanned by the rows of $\mathbf{SA}$ (and $(\cdot)^\dagger$ denotes the Moore-Penrose pseudoinverse). For this reason, the quantity that has appeared ubiquitously in the error analysis of RandNLA sketching is what we call the residual projection matrix:

$$\text{(residual projection matrix)} \quad \mathbf{P}_\perp := \mathbf{I} - \mathbf{P} = \mathbf{I} - (\mathbf{SA})^\dagger \mathbf{SA}.$$

Since $\mathbf{P}_\perp$ is random, the average performance of the sketch can often be characterized by its expectation, $\mathbb{E}[\mathbf{P}_\perp]$. For example, the low-rank approximation error of the sketch can be expressed as $\mathbb{E}[\|\mathbf{A} - \mathbf{AP}\|_F^2] = \operatorname{tr} \mathbf{A}^\top \mathbf{A} \, \mathbb{E}[\mathbf{P}_\perp]$, where $\|\cdot\|_F$ denotes the Frobenius norm. A similar formula follows for the trace norm error of a sketched Nyström approximation [WS01, GM16]. Among others, this approximation error appears in the analysis of sketched kernel ridge regression [FSS20] and Gaussian process regression [BRVDW19]. Furthermore, a variety of iterative algorithms, such as randomized second-order methods for convex optimization [QRTF16, QR16, GKLR19, GRB20] and linear system solvers based on the generalized Kaczmarz method [GR15], have convergence guarantees which depend on the extreme eigenvalues of $\mathbb{E}[\mathbf{P}_\perp]$. Finally, a generalized form of the expected residual projection has been recently used to model the implicit regularization of the interpolating solutions in over-parameterized linear models [DLM19, BLLT19].

## 1.1 Main result

Despite its prevalence in the literature, the expected residual projection is not well understood, even in such simple cases as when $\mathbf{S}$ is a Gaussian sketch (i.e., with i.i.d. standard normal entries). We address this by providing a surrogate expression, i.e., a simple analytically tractable approximation, for this matrix quantity:

$$\mathbb{E}[\mathbf{P}_\perp] \overset{\epsilon}{\simeq} \bar{\mathbf{P}}_\perp := (\gamma \mathbf{A}^\top \mathbf{A} + \mathbf{I})^{-1}, \quad \text{with } \gamma > 0 \text{ s.t. } \operatorname{tr} \bar{\mathbf{P}}_\perp = n - k. \tag{1}$$

Here, $\overset{\epsilon}{\simeq}$ means that while the surrogate expression is not exact, it approximates the true quantity up to some $\epsilon$ accuracy. Our main result provides a rigorous approximation guarantee for this surrogate expression with respect to a range of sketching matrices $\mathbf{S}$, including the standard Gaussian and Rademacher sketches. We state the result using the positive semi-definite ordering denoted by $\preceq$.

**Theorem 1.** *Let $\mathbf{S}$ be a sketch of size $k$ with i.i.d. mean-zero sub-gaussian entries and let $r = \|\mathbf{A}\|_F^2 / \|\mathbf{A}\|^2$ be the stable rank of $\mathbf{A}$. If we let $\rho = r/k$ be a fixed constant larger than $1$, then*

$$(1 - \epsilon) \bar{\mathbf{P}}_\perp \preceq \mathbb{E}[\mathbf{P}_\perp] \preceq (1 + \epsilon) \bar{\mathbf{P}}_\perp \quad \text{for} \quad \epsilon = O(\tfrac{1}{\sqrt{r}}).$$

In other words, when the sketch size $k$ is smaller than the stable rank $r$ of $\mathbf{A}$, then the discrepancy between our surrogate expression $\bar{\mathbf{P}}_\perp$ and $\mathbb{E}[\mathbf{P}_\perp]$ is of the order $1/\sqrt{r}$, where the big-O notation hides only the dependence on $\rho$ and on the sub-gaussian constant (see Theorem 2 for more details). Our proof of Theorem 1 is inspired by the techniques from random matrix theory which have been used to analyze the asymptotic spectral distribution of large random matrices by focusing on the associated matrix resolvents and Stieltjes transforms [HLN+07, BS10]. However, our analysis is novel in several respects:

1. The residual projection matrix can be obtained from the appropriately scaled resolvent matrix $z(\mathbf{A}^\top \mathbf{S}^\top \mathbf{SA} + z\mathbf{I})^{-1}$ by taking $z \to 0$. Prior work (e.g., [HMRT19]) combined this with an exchange-of-limits argument to analyze the asymptotic behavior of the residual projection. This approach, however, does not allow for a precise control in finite-dimensional problems. We are able to provide a more fine-grained, non-asymptotic analysis by working directly with the residual projection itself, instead of the resolvent.

2. We require no assumptions on the largest and smallest singular value of $\mathbf{A}$. Instead, we derive our bounds in terms of the stable rank of $\mathbf{A}$ (as opposed to its actual rank), which implicitly compensates for ill-conditioned data matrices.

3. We obtain upper/lower bounds for $\mathbb{E}[\mathbf{P}_\perp]$ in terms of the positive semi-definite ordering $\preceq$, which can be directly converted to guarantees for the precise expressions of expected low-rank approximation error derived in the following section.

It is worth mentioning that the proposed analysis is significantly different from the sketching literature based on subspace embeddings (e.g., [Sar06, CW17, NN13, CEM+15, CNW16]), in the sense that here our object of interest is not to obtain a worst-case approximation with high probability, but rather, our analysis provides *precise* characterization on the *expected* residual projection matrix that goes *beyond worst-case bounds*. From an application perspective, the subspace embedding property is neither sufficient nor necessary for many numerical implementations of sketching [AMT10, MSM14], or statistical results [RM16, DL19, YLDW20], as well as in the context of iterative optimization and implicit regularization (see Sections 1.3 and 1.4 below), which are discussed in detail as concrete applications of the proposed analysis.

## 1.2 Low-rank approximation

We next provide some immediate corollaries of Theorem 1, where we use $x \overset{\epsilon}{\simeq} y$ to denote a multiplicative approximation $|x - y| \leq \epsilon y$. Note that our analysis is new even for the classical Gaussian sketch where the entries of $\mathbf{S}$ are i.i.d. standard normal. However the results apply more broadly, including a standard class of data-base friendly Rademacher sketches where each entry $s_{ij}$ is a $\pm 1$ Rademacher random variable [Ach03]. We start by analyzing the Frobenius norm error $\|\mathbf{A} - \mathbf{AP}\|_F^2 = \operatorname{tr} \mathbf{A}^\top \mathbf{A} \, \mathbf{P}_\perp$ of sketched low-rank approximations. Note that by the definition of $\gamma$ in (1), we have $k = \operatorname{tr}(\mathbf{I} - \bar{\mathbf{P}}_\perp) = \operatorname{tr} \gamma \mathbf{A}^\top \mathbf{A}(\gamma \mathbf{A}^\top \mathbf{A} + \mathbf{I})^{-1}$, so the surrogate expression we obtain for the expected error is remarkably simple.

**Corollary 1.** *Let $\sigma_i$ be the singular values of $\mathbf{A}$. Under the assumptions of Theorem 1, we have:*

$$\mathbb{E}\big[\|\mathbf{A} - \mathbf{AP}\|_F^2\big] \overset{\epsilon}{\simeq} k/\gamma \quad \text{for } \gamma > 0 \text{ s.t. } \sum_i \frac{\gamma \sigma_i^2}{\gamma \sigma_i^2 + 1} = k.$$

**Remark 1.** *The parameter $\gamma = \gamma(k)$ increases at least linearly as a function of $k$, which is why the expected error will always decrease with increasing $k$. For example, when the singular values of $\mathbf{A}$ exhibit exponential decay, i.e., $\sigma_i^2 = C \cdot \alpha^{i-1}$ for $\alpha \in (0, 1)$, then the error also decreases exponentially, at the rate of $k/(\alpha^{-k} - 1)$. We discuss this further in Section 3, giving explicit formulas for the error as a function of $k$ under both exponential and polynomial spectral decay profiles.*

The above result is important for many RandNLA methods, and it is also relevant in the context of kernel methods, where the data is represented via a positive semi-definite $m \times m$ kernel matrix $\mathbf{K}$ which corresponds to the matrix of dot-products of the data vectors in some reproducible kernel Hilbert space. In this context, sketching can be applied directly to the matrix $\mathbf{K}$ via an extended variant of the Nyström method [GM16]. A Nyström approximation constructed from a sketching matrix $\mathbf{S}$ is defined as $\tilde{\mathbf{K}} = \mathbf{C}^\top \mathbf{W}^\dagger \mathbf{C}$, where $\mathbf{C} = \mathbf{SK}$ and $\mathbf{W} = \mathbf{SKS}^\top$, and it is applicable to a variety of settings, including Gaussian Process regression, kernel machines and Independent Component Analysis [BRVDW19, WS01, BJ03]. By setting $\mathbf{A} = \mathbf{K}^{\frac{1}{2}}$, it is easy to see [DKM20] that the trace norm error $\|\mathbf{K} - \tilde{\mathbf{K}}\|_*$ is identical to the squared Frobenius norm error of the low-rank sketch $\mathbf{SA}$, so Corollary 1 implies that

$$\mathbb{E}\big[\|\mathbf{K} - \tilde{\mathbf{K}}\|_*\big] \overset{\epsilon}{\simeq} k/\gamma \quad \text{for } \gamma > 0 \text{ s.t. } \sum_i \frac{\gamma \lambda_i}{\gamma \lambda_i + 1} = k, \tag{2}$$

with any sub-gaussian sketch, where $\lambda_i$ denote the eigenvalues of $\mathbf{K}$. Our error analysis given in Section 3 is particularly relevant here, since commonly used kernels such as the Radial Basis Function (RBF) or the Matérn kernel induce a well-understood eigenvalue decay [SZW+97, RW06].

Metrics other than the aforementioned Frobenius norm error, such as the spectral norm error [HMT11], are also of significant interest in the low-rank approximation literature. We leave these directions for future investigation.

## 1.3 Randomized iterative optimization

We next turn to a class of iterative methods which take advantage of sketching to reduce the per iteration cost of optimization. These methods have been developed in a variety of settings, from solving linear systems to convex optimization and empirical risk minimization, and in many cases the residual projection matrix appears as a black box quantity whose spectral properties determine the convergence behavior of the algorithms [GR15]. With our new results, we can precisely characterize not only the rate of convergence, but also, in some cases, the complete evolution of the parameter vector, for the following algorithms:

1. *Generalized Kaczmarz method* [GR15] for approximately solving a linear system $\mathbf{A}\mathbf{x} = \mathbf{b}$;
2. *Randomized Subspace Newton* [GKLR19], a second order method, where we sketch the Hessian matrix.
3. *Jacobian Sketching* [GRB20], a class of first order methods which use additional information via a weight matrix $\mathbf{W}$ that is sketched at every iteration.

We believe that extensions of our techniques will apply to other algorithms, such as that of [LPP19].

We next give a result in the context of linear systems for the generalized Kaczmarz method [GR15], but a similar convergence analysis is given for the methods of [GKLR19, GRB20] in Appendix B.

**Corollary 2.** *Let $\mathbf{x}^*$ be the unique solution of $\mathbf{A}\mathbf{x}^* = \mathbf{b}$ and consider the iterative algorithm:*

$$\mathbf{x}^{t+1} = \underset{\mathbf{x}}{\arg\min} \|\mathbf{x} - \mathbf{x}^t\|^2 \quad \text{subject to} \quad \mathbf{S}\mathbf{A}\mathbf{x} = \mathbf{S}\mathbf{b}.$$

*Under the assumptions of Theorem 1, with $\gamma$ defined in (1) and $r = \|\mathbf{A}\|_F^2/\|\mathbf{A}\|^2$, we have:*

$$\mathbb{E}\big[\mathbf{x}^{t+1} - \mathbf{x}^*\big] \overset{\epsilon}{\simeq} (\gamma\mathbf{A}^\top\mathbf{A} + \mathbf{I})^{-1}\, \mathbb{E}\big[\mathbf{x}^t - \mathbf{x}^*\big] \quad \text{for} \quad \epsilon = O(\tfrac{1}{\sqrt{r}}).$$

The corollary follows from Theorem 1 combined with Theorem 4.1 in [GR15]. Note that when $\mathbf{A}^\top\mathbf{A}$ is positive definite then $(\gamma\mathbf{A}^\top\mathbf{A} + \mathbf{I})^{-1} \prec \mathbf{I}$, so the algorithm will converge from any starting point, and the worst-case convergence rate of the above method can be obtained by evaluating the largest eigenvalue of $(\gamma\mathbf{A}^\top\mathbf{A} + \mathbf{I})^{-1}$. However the result itself is much stronger, in that it can be used to describe the (expected) trajectory of the iterates for any starting point $\mathbf{x}^0$. Moreover, when the spectral decay profile of $\mathbf{A}$ is known, then the explicit expressions for $\gamma$ as a function of $k$ derived in Section 3 can be used to characterize the convergence properties of generalized Kaczmarz as well as other methods discussed above.

## 1.4 Implicit regularization

Setting $\mathbf{x}^t = \mathbf{0}$, we can view one step of the iterative method in Corollary 2 as finding a minimum norm interpolating solution of an under-determined linear system $(\mathbf{S}\mathbf{A}, \mathbf{S}\mathbf{b})$. Recent interest in the generalization capacity of over-parameterized machine learning models has motivated extensive research on the statistical properties of such interpolating solutions [e.g., BLLT19, HMRT19, DLM19]. In this context, Theorem 1 provides new evidence for the implicit regularization conjecture posed by [DLM19] (see their Theorem 2 and associated discussion), with the amount of regularization equal $\frac{1}{\gamma}$, where $\gamma$ is implicitly defined in (1):

$$\underbrace{\mathbb{E}\left[\underset{\mathbf{x}}{\arg\min}\|\mathbf{x}\|^2 \ \text{s.t.} \ \mathbf{S}\mathbf{A}\mathbf{x} = \mathbf{S}\mathbf{b}\right] - \mathbf{x}^*}_{\text{Bias of sketched minimum norm solution}} \overset{\epsilon}{\simeq} \underbrace{\underset{\mathbf{x}}{\arg\min}\left\{\|\mathbf{A}\mathbf{x} - \mathbf{b}\|^2 + \tfrac{1}{\gamma}\|\mathbf{x}\|^2\right\} - \mathbf{x}^*}_{\text{Bias of } l_2\text{-regularized solution}}.$$

While implicit regularization has received attention recently in the context of SGD algorithms for overparameterized machine learning models, it was originally discussed in the context of approximation algorithms more generally [Mah12]. Recent work has made precise this notion in the context of RandNLA [DLM19], and our results here can be viewed in terms of implicit regularization of scalable RandNLA methods.

## 1.5 Related work

A significant body of research has been dedicated to understanding the guarantees for low-rank approximation via sketching, particularly in the context of RandNLA [DM16, DM18]. This line of work includes i.i.d. row sampling methods [BMD08, AM15] which preserve the structure of the data, and data-oblivious methods such as Gaussian and Rademacher sketches [Mah11, HMT11, Woo14]. However, all of these results focus on worst-case upper bounds on the approximation error. One exception is a recent line of works on non-i.i.d. row sampling with Determinantal Point Processes (DPP, [DM20]). In this case, exact analysis of the low-rank approximation error [DKM20], as well as precise convergence analysis of stochastic second order methods [MDK20], have been obtained. Remarkably, the expressions they obtain are analogous to (1), despite using completely different techniques. However, their analysis is limited only to DPP-based sketches, which are

considerably more expensive to construct and thus much less widely used. The connection between DPPs and Gaussian sketches was recently explored by [DLM19] in the context of analyzing the implicit regularization effect of choosing a minimum norm solution in under-determined linear regression. They conjectured that the expectation formulas obtained for DPPs are a good proxy for the corresponding quantities obtained under a Gaussian distribution. Similar observations were made by [DBPM20] in the context of sketching for regularized least squares and second order optimization. While both of these works only provide empirical evidence for this particular claim, our Theorem 1 can be viewed as the first theoretical non-asymptotic justification of that conjecture.

The effectiveness of sketching has also been extensively studied in the context of second order optimization. These methods differ depending on how the sketch is applied to the Hessian matrix, and whether or not it is applied to the gradient as well. The class of methods discussed in Section 1.3, including Randomized Subspace Newton and the Generalized Kaczmarz method, relies on projecting the Hessian downto a low-dimensional subspace, which makes our results directly applicable. A related family of methods uses the so-called Iterative Hessian Sketch (IHS) approach [PW16, LP19]. The similarities between IHS and the Subspace Newton-type methods (see [QRTF16] for a comparison) suggest that our techniques could be extended to provide precise convergence guarantees also to the IHS. Finally, yet another family of Hessian sketching methods has been studied by [RKM19, WGM17, XRKM17, YXRKM18, RLXM18, WRXM17, DM19]. These methods preserve the rank of the Hessian, and thus they provide somewhat different convergence guarantees that do not rely on the residual projection matrix.

# 2 Precise analysis of the residual projection

In this section, we give a detailed statement of our main technical result, along with a sketch of the proof. First, recall the definition of sub-gaussian random variables and vectors.

**Definition 1.** *We say that $x$ is a $K$-sub-gaussian random variable if its sub-gaussian Orlicz norm $\|x\|_{\psi_2} \leq K$, where $\|x\|_{\psi_2} := \inf\{t > 0 : \mathbb{E}[\exp(x^2/t^2)] \leq 2\}$. Similarly, we say that a random vector $\mathbf{x}$ is $K$-sub-gaussian if for all $\|\mathbf{a}\| \leq 1$ we have $\|\mathbf{x}^\top \mathbf{a}\|_{\psi_2} \leq K$.*

For convenience, we state the main result in a slightly different form than Theorem 1. Namely, we replace the $m \times n$ matrix $\mathbf{A}$ with a positive semi-definite $n \times n$ matrix $\mathbf{\Sigma}^{\frac{1}{2}}$. Furthermore, instead of a sketch $\mathbf{S}$ with i.i.d. sub-gaussian entries, we use a random matrix $\mathbf{Z}$ with i.i.d. sub-gaussian *rows*, which is a strictly weaker condition because it allows for the entries of each row to be correlated. Since the rows of $\mathbf{Z}$ are also assumed to have mean zero and identity covariance, each row of $\mathbf{Z}\mathbf{\Sigma}^{\frac{1}{2}}$ has covariance $\mathbf{\Sigma}$. In Section 2.2 we show how to convert this statement back to the form of Theorem 1.

**Theorem 2.** *Let $\mathbf{P}_\perp = \mathbf{I} - \mathbf{X}^\dagger \mathbf{X}$ for $\mathbf{X} = \mathbf{Z}\mathbf{\Sigma}^{\frac{1}{2}}$, where $\mathbf{Z} \in \mathbb{R}^{k \times n}$ has i.i.d. $K$-sub-gaussian rows with zero mean and identity covariance, and $\mathbf{\Sigma}$ is an $n \times n$ positive semi-definite matrix. Define:*

$$\bar{\mathbf{P}}_\perp = (\gamma\mathbf{\Sigma} + \mathbf{I})^{-1}, \quad \text{such that} \quad \operatorname{tr}\bar{\mathbf{P}}_\perp = n - k.$$

*Let $r = \operatorname{tr}(\mathbf{\Sigma})/\|\mathbf{\Sigma}\|$ be the stable rank of $\mathbf{\Sigma}^{\frac{1}{2}}$ and fix $\rho = r/k > 1$. There exists a constant $C_\rho > 0$, depending only on $\rho$ and $K$, such that if $r \geq C_\rho$, then*

$$\left(1 - \frac{C_\rho}{\sqrt{r}}\right) \cdot \bar{\mathbf{P}}_\perp \preceq \mathbb{E}[\mathbf{P}_\perp] \preceq \left(1 + \frac{C_\rho}{\sqrt{r}}\right) \cdot \bar{\mathbf{P}}_\perp. \tag{3}$$

We first provide the following informal derivation of the expression for $\bar{\mathbf{P}}_\perp$ given in Theorem 2. Let us use $\mathbf{P}$ to denote the matrix $\mathbf{X}^\dagger \mathbf{X} = \mathbf{I} - \mathbf{P}_\perp$. Using a rank-one update formula for the Moore-Penrose pseudoinverse (see Lemma 1 in the appendix) we have

$$\mathbf{I} - \mathbb{E}[\mathbf{P}_\perp] = \mathbb{E}[\mathbf{P}] = \mathbb{E}\big[(\mathbf{X}^\top\mathbf{X})^\dagger\mathbf{X}^\top\mathbf{X}\big] = \sum_{i=1}^{k}\mathbb{E}[(\mathbf{X}^\top\mathbf{X})^\dagger\mathbf{x}_i\mathbf{x}_i^\top] = k\,\mathbb{E}\left[\frac{(\mathbf{I} - \mathbf{P}_{-k})\mathbf{x}_k\mathbf{x}_k^\top}{\mathbf{x}_k^\top(\mathbf{I} - \mathbf{P}_{-k})\mathbf{x}_k}\right],$$

where we use $\mathbf{x}_i^\top$ to denote the $i$-th row of $\mathbf{X}$, and $\mathbf{P}_{-k} = \mathbf{X}_{-k}^\dagger\mathbf{X}_{-k}$, where $\mathbf{X}_{-i}$ is the matrix $\mathbf{X}$ without its $i$-th row. Due to the sub-gaussianity of $\mathbf{x}_k$, the quadratic form $\mathbf{x}_k^\top(\mathbf{I} - \mathbf{P}_{-k})\mathbf{x}_k$ in the denominator concentrates around its expectation (with respect to $\mathbf{x}_k$), i.e., $\operatorname{tr}\mathbf{\Sigma}(\mathbf{I} - \mathbf{P}_{-k})$, where we

use $\mathbb{E}[\mathbf{x}_k \mathbf{x}_k^\top] = \boldsymbol{\Sigma}$. Further note that, with $\mathbf{P}_{-k} \simeq \mathbf{P}$ for large $k$ and $\frac{1}{k}\mathrm{tr}\boldsymbol{\Sigma}(\mathbf{I} - \mathbf{P}_{-k}) \simeq \frac{1}{k}\mathrm{tr}\boldsymbol{\Sigma}\mathbb{E}[\mathbf{P}_\perp]$ from a concentration argument, we conclude that

$$\mathbf{I} - \mathbb{E}[\mathbf{P}_\perp] \simeq \frac{k\mathbb{E}[\mathbf{P}_\perp]\boldsymbol{\Sigma}}{\mathrm{tr}\boldsymbol{\Sigma}\mathbb{E}[\mathbf{P}_\perp]} \qquad \Longrightarrow \qquad \mathbb{E}[\mathbf{P}_\perp] \simeq \left(\frac{k\boldsymbol{\Sigma}}{\mathrm{tr}\boldsymbol{\Sigma}\mathbb{E}[\mathbf{P}_\perp]} + \mathbf{I}\right)^{-1},$$

and thus $\mathbb{E}[\mathbf{P}_\perp] \simeq \bar{\mathbf{P}}_\perp$ for $\bar{\mathbf{P}}_\perp = (\gamma\boldsymbol{\Sigma} + \mathbf{I})^{-1}$ and $\gamma^{-1} = \frac{1}{k}\mathrm{tr}\boldsymbol{\Sigma}\bar{\mathbf{P}}_\perp$. This leads to the (implicit) expression for $\bar{\mathbf{P}}_\perp$ and $\gamma$ given in Theorem 2.

## 2.1 Proof sketch of Theorem 2

To make the above intuition rigorous, we next present a proof sketch for Theorem 2, with the detailed proof deferred to Appendix A. The proof can be divided into the following three steps.

**Step 1.** First note that, to obtain the lower and upper bound for $\mathbb{E}[\mathbf{P}_\perp]$ in the sense of symmetric matrix as in Theorem 2, it suffices to bound the spectral norm $\|\mathbf{I} - \mathbb{E}[\mathbf{P}_\perp]\bar{\mathbf{P}}_\perp^{-1}\| \leq \frac{C_\rho}{\sqrt{r}}$, so that, with $\frac{\rho-1}{\rho}\mathbf{I} \preceq \bar{\mathbf{P}}_\perp \preceq \mathbf{I}$ for $\rho = r/k > 1$ from the definition of $\bar{\mathbf{P}}_\perp$, we have

$$\|\mathbf{I} - \bar{\mathbf{P}}_\perp^{-\frac{1}{2}}\mathbb{E}[\mathbf{P}_\perp]\bar{\mathbf{P}}_\perp^{-\frac{1}{2}}\| = \|\bar{\mathbf{P}}_\perp^{-\frac{1}{2}}(\mathbf{I} - \mathbb{E}[\mathbf{P}_\perp]\bar{\mathbf{P}}^{-1})\bar{\mathbf{P}}_\perp^{\frac{1}{2}}\| \leq \frac{C_\rho}{\sqrt{r}}\sqrt{\frac{\rho}{\rho-1}} =: \epsilon.$$

This means that all eigenvalues of the p.s.d. matrix $\bar{\mathbf{P}}_\perp^{-\frac{1}{2}}\mathbb{E}[\mathbf{P}_\perp]\bar{\mathbf{P}}_\perp^{-\frac{1}{2}}$ lie in the interval $[1 - \epsilon, 1 + \epsilon]$, so $(1-\epsilon)\mathbf{I} \preceq \bar{\mathbf{P}}_\perp^{-\frac{1}{2}}\mathbb{E}[\mathbf{P}_\perp]\bar{\mathbf{P}}_\perp^{-\frac{1}{2}} \preceq (1+\epsilon)\mathbf{I}$. Multiplying by $\bar{\mathbf{P}}_\perp^{\frac{1}{2}}$ from both sides, we obtain the desired bound.

**Step 2.** Then, we carefully design an event $E$ that (i) is provable to occur with high probability and (ii) ensures that the denominators in the following decomposition are bounded away from zero:

$$\mathbf{I} - \mathbb{E}[\mathbf{P}_\perp]\bar{\mathbf{P}}_\perp^{-1} = \mathbb{E}[\mathbf{P}] - \gamma\mathbb{E}[\mathbf{P}_\perp]\boldsymbol{\Sigma} = \mathbb{E}[\mathbf{P} \cdot \mathbf{1}_E] + \mathbb{E}[\mathbf{P} \cdot \mathbf{1}_{\neg E}] - \gamma\mathbb{E}[\mathbf{P}_\perp]\boldsymbol{\Sigma}$$

$$= \gamma \underbrace{\mathbb{E}\left[(\bar{s} - \hat{s}) \cdot \frac{(\mathbf{I} - \mathbf{P}_{-k})\mathbf{x}_k\mathbf{x}_k^\top}{\mathbf{x}_k^\top(\mathbf{I} - \mathbf{P}_{-k})\mathbf{x}_k} \cdot \mathbf{1}_E\right]}_{\mathbf{T}_1} - \gamma\underbrace{\mathbb{E}[(\mathbf{I} - \mathbf{P}_{-k})\mathbf{x}_k\mathbf{x}_k^\top \cdot \mathbf{1}_{\neg E}]}_{\mathbf{T}_2}$$

$$+ \gamma\underbrace{\mathbb{E}[\mathbf{P} - \mathbf{P}_{-k}]\boldsymbol{\Sigma}}_{\mathbf{T}_3} + \underbrace{\mathbb{E}[\mathbf{P} \cdot \mathbf{1}_{\neg E}]}_{\mathbf{T}_4},$$

where we let $\hat{s} = \mathbf{x}_k^\top(\mathbf{I} - \mathbf{P}_{-k})\mathbf{x}_k$ and $\bar{s} = k/\gamma$.

**Step 3.** It then remains to bound the spectral norms of $\mathbf{T}_1, \mathbf{T}_2, \mathbf{T}_3, \mathbf{T}_4$ respectively to reach the conclusion. More precisely, the terms $\|\mathbf{T}_2\|$ and $\|\mathbf{T}_4\|$ are proportional to $\mathrm{Pr}(\neg E)$, while the term $\|\mathbf{T}_3\|$ can be bounded using the rank-one update formula for the pseudoinverse (Lemma 1 in the appendix). The remaining term $\|\mathbf{T}_1\|$ is more subtle and can be bounded with a careful application of the Hanson-Wright type [RV13] sub-gaussian concentration inequalities (Lemmas 2 and 3 in the appendix). This allows for a bound on the operator norm $\|\mathbf{I} - \mathbb{E}[\mathbf{P}_\perp]\bar{\mathbf{P}}_\perp^{-1}\|$ and hence the conclusion.

## 2.2 Proof of Theorem 1

We now discuss how Theorem 1 can be obtained from Theorem 2. The crucial difference between the statements is that in Theorem 1 we let $\mathbf{A}$ be an arbitrary rectangular matrix, whereas in Theorem 2 we instead use a square, symmetric and positive semi-definite matrix $\boldsymbol{\Sigma}$. To convert between the two notations, consider the SVD decomposition $\mathbf{A} = \mathbf{U}\mathbf{D}\mathbf{V}^\top$ of $\mathbf{A} \in \mathbb{R}^{m \times n}$ (recall that we assume $m \geq n$), where $\mathbf{U} \in \mathbb{R}^{m \times n}$ and $\mathbf{V} \in \mathbb{R}^{n \times n}$ have orthonormal columns and $\mathbf{D}$ is a diagonal matrix. Now, let $\mathbf{Z} = \mathbf{S}\mathbf{U}$, $\boldsymbol{\Sigma} = \mathbf{D}^2$ and $\mathbf{X} = \mathbf{Z}\boldsymbol{\Sigma}^{\frac{1}{2}} = \mathbf{S}\mathbf{U}\mathbf{D}$. Using the fact that $\mathbf{V}^\top\mathbf{V} = \mathbf{V}\mathbf{V}^\top = \mathbf{I}$, it follows that:

$$\mathbf{I} - (\mathbf{S}\mathbf{A})^\dagger\mathbf{S}\mathbf{A} = \mathbf{V}(\mathbf{I} - \mathbf{X}^\dagger\mathbf{X})\mathbf{V}^\top \quad \text{and} \quad (\gamma\mathbf{A}^\top\mathbf{A} + \mathbf{I})^{-1} = \mathbf{V}(\gamma\boldsymbol{\Sigma} + \mathbf{I})^{-1}\mathbf{V}^\top.$$

Note that since $\|\mathbf{U}\mathbf{v}\| = \|\mathbf{v}\|$, the rows of $\mathbf{Z}$ are sub-gaussian with the same constant as the rows of $\mathbf{S}$. Moreover, using the fact that $\mathbf{B} \preceq \mathbf{C}$ implies $\mathbf{V}\mathbf{B}\mathbf{V}^\top \preceq \mathbf{V}\mathbf{C}\mathbf{V}^\top$ for any p.s.d. matrices $\mathbf{B}$ and $\mathbf{C}$, Theorem 1 follows as a corollary of Theorem 2.

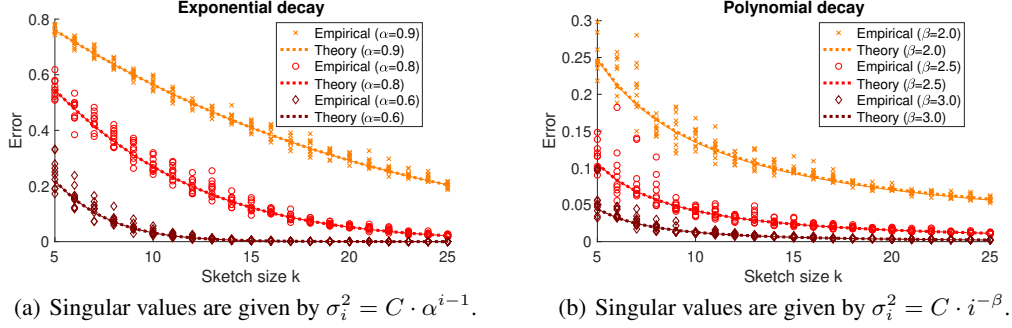

(a) Singular values are given by $\sigma_i^2 = C \cdot \alpha^{i-1}$.  (b) Singular values are given by $\sigma_i^2 = C \cdot i^{-\beta}$.

Figure 1: Theoretical predictions of low-rank approximation error of a Gaussian sketch under known spectral decays, compared to the empirical results. The constant $C$ is scaled so that $\|\mathbf{A}\|_F^2 = 1$ and we let $n = m = 1000$. For the theory, we plot the explicit formulas (5) and (6) (dashed lines), as well as the implicit expression from Corollary 1 (thin solid lines) obtained by numerically solving (4). Observe that the explicit and implicit predictions are nearly (but not exactly) identical.

## 3  Explicit formulas under known spectral decay

The expression we give for the expected residual projection, $\mathbb{E}[\mathbf{P}_\perp] \simeq (\gamma \mathbf{A}^\top \mathbf{A} + \mathbf{I})^{-1}$, is implicit in that it depends on the parameter $\gamma$ which is the solution of the following equation:

$$\sum_{i \geq 1} \frac{\gamma \sigma_i^2}{\gamma \sigma_i^2 + 1} = k, \qquad \text{where } \sigma_i \text{ are the singular values of } \mathbf{A}. \tag{4}$$

In general, it is impossible to solve this equation analytically, i.e., to write $\gamma$ as an explicit formula of $n$, $k$ and the singular values of $\mathbf{A}$. However, we show that when the singular values exhibit a known rate of decay, then it is possible to obtain explicit formulas for $\gamma$. In particular, this allows us to provide precise and easily interpretable rates of decay for the low-rank approximation error of a sub-gaussian sketch.

Matrices that have known spectral decay, most commonly with either exponential or polynomial rate, arise in many machine learning problems [MDK20]. Such behavior can be naturally occurring in data, or it can be induced by feature expansion using, say, the RBF kernel (for exponential decay) [SZW+97] or the Matérn kernel (for polynomial decay) [RW06]. Understanding these two classes of decay plays an important role in distinguishing the properties of light-tailed and heavy-tailed data distributions. Note that in the kernel setting we may often represent our data via the $m \times m$ kernel matrix $\mathbf{K}$, instead of the $m \times n$ data matrix $\mathbf{A}$, and study the sketched Nyström method [GM16] for low-rank approximation. To handle the kernel setting in our analysis, it suffices to replace the squared singular values $\sigma_i^2$ of $\mathbf{A}$ with the eigenvalues of $\mathbf{K}$.

### 3.1  Exponential spectral decay

Suppose that the squared singular values of $\mathbf{A}$ exhibit exponential decay, i.e. $\sigma_i^2 = C \cdot \alpha^{i-1}$, where $C$ is a constant and $\alpha \in (0, 1)$. For simplicity of presentation, we will let $m, n \to \infty$. Under this spectral decay, we can approximate the sum in (4) by the analytically computable integral $\int_y^\infty \frac{1}{1 + (C\gamma)^{-1}\alpha^{-x}} dx$, obtaining $\gamma \approx (\alpha^{-k} - 1)\sqrt{\alpha}/C$. Applying this to the formula from Corollary 1, we can express the low-rank approximation error for a sketch of size $k$ as follows:

$$\mathbb{E}\big[\|\mathbf{A} - \mathbf{AP}\|_F^2\big] \approx \frac{C}{\sqrt{\alpha}} \cdot \frac{k}{\alpha^{-k} - 1}, \quad \text{when} \quad \sigma_i^2 = C \cdot \alpha^{i-1} \text{ for all } i. \tag{5}$$

In Figure 1a, we plot the above formula against the numerically obtained implicit expression from Corollary 1, as well as empirical results for a Gaussian sketch. First, we observe that the theoretical predictions closely align with empirical values even after the sketch size crosses the stable rank $r \approx \frac{1}{1-\alpha}$, suggesting that Theorem 1 can be extended to this regime. Second, while it is not surprising that the error decays at a similar rate as the singular values, our predictions offer a much more precise description, down to lower order effects and even constant factors. For instance, we observe that the

error (normalized by $\|\mathbf{A}\|_F^2$, as in the figure) only starts decaying exponentially after $k$ crosses the stable rank, and until that point it decreases at a linear rate with slope $-\frac{1-\alpha}{2\sqrt{\alpha}}$.

## 3.2 Polynomial spectral decay

We now turn to polynomial spectral decay, which is a natural model for analyzing heavy-tailed data distributions. Let $\mathbf{A}$ have squared singular values $\sigma_i^2 = C \cdot i^{-\beta}$ for some $\beta \geq 2$, and let $m, n \to \infty$. As in the case of exponential decay, we use the integral $\int_y^\infty \frac{1}{1+(C\gamma)^{-1}x^{-\beta}}dx$ to approximate the sum in (4), and solve for $\gamma$, obtaining $\gamma \approx \left((k+\frac{1}{2})\frac{\beta}{\pi}\sin(\frac{\pi}{\beta})\right)^\beta$. Combining this with Corollary 1 we get:

$$\mathbb{E}\big[\|\mathbf{A} - \mathbf{AP}\|_F^2\big] \approx C \cdot \frac{k}{(k+\frac{1}{2})^\beta}\left(\frac{\pi/\beta}{\sin(\pi/\beta)}\right)^\beta, \quad \text{when} \quad \sigma_i^2 = C \cdot i^{-\beta} \text{ for all } i. \tag{6}$$

Figure 1b compares our predictions to the empirical results for several values of $\beta$. In all of these cases, the stable rank is close to 1, and yet the theoretical predictions align very well with the empirical results. Overall, the asymptotic rate of decay of the error is $k^{1-\beta}$. However it is easy to verify that the lower order effect of $(k+\frac{1}{2})^\beta$ appearing instead of $k^\beta$ in (6) significantly changes the trajectory for small values of $k$. Also, note that as $\beta$ grows large, the constant $\left(\frac{\pi/\beta}{\sin(\pi/\beta)}\right)^\beta$ goes to 1, but it plays a significant role for $\beta = 2$ or 3 (roughly, scaling the expression by a factor of 2). Finally, we remark that for $\beta \in (1, 2)$, our integral approximation of (4) becomes less accurate. We expect that a corrected expression is possible, but likely more complicated and less interpretable.

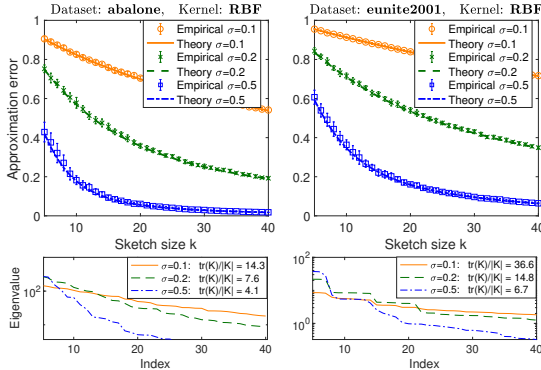

Figure 2: Theoretical predictions versus approximation error for the sketched Nyström with the RBF kernel (spectral decay shown at the bottom).

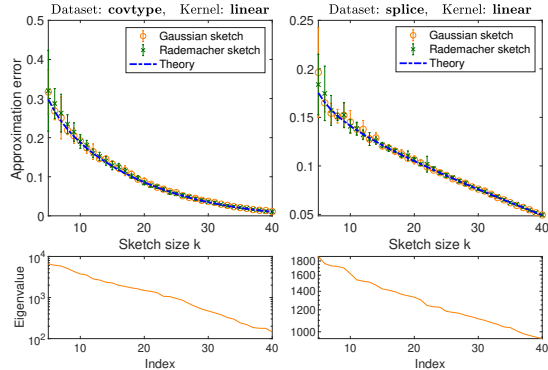

Figure 3: Theoretical predictions versus approximation error for the Gaussian and Rademacher sketches (spectral decay shown at the bottom).

## 4 Empirical results

In this section, we numerically verify the accuracy of our theoretical predictions for the low-rank approximation error of sketching on benchmark datasets from the libsvm repository [CL11] (further numerical results are in Appendix C). We repeated every experiment 10 times, and plot both the average and standard deviation of the results. We use the following $k \times m$ sketching matrices $\mathbf{S}$:

1. *Gaussian sketch:* with i.i.d. standard normal entries;
2. *Rademacher sketch:* with i.i.d. entries equal 1 with probability 0.5 and $-1$ otherwise.

**Varying spectral decay.** To demonstrate the role of spectral decay and the stable rank on the approximation error, we performed feature expansion using the radial basis function (RBF) kernel $k(\mathbf{a}_i, \mathbf{a}_j) = \exp(-\|\mathbf{a}_i - \mathbf{a}_j\|^2/(2\sigma^2))$, obtaining an $m \times m$ kernel matrix $\mathbf{K}$. We used the sketched Nyström method to construct a low-rank approximation $\tilde{\mathbf{K}} = \mathbf{KS}^\top(\mathbf{SKS}^\top)^\dagger\mathbf{SK}$, and computed the normalized trace norm error $\|\mathbf{K} - \tilde{\mathbf{K}}\|_*/\|\mathbf{K}\|_*$. The theoretical predictions are coming from (2), which in turn uses Theorem 1. Following [GM16], we use the RBF kernel because varying the scale parameter $\sigma$ allows us to observe the approximation error under qualitatively different spectral decay

profiles of the kernel. In Figure 2, we present the results for the Gaussian sketch on two datasets, with three values of $\sigma$, and in all cases our theory aligns with the empirical results. Furthermore, as smaller $\sigma$ leads to slower spectral decay and larger stable rank, it also makes the approximation error decay more linearly for small sketch sizes. This behavior is predicted by our explicit expressions (5) for the error under exponential spectral decay from Section 3. Once the sketch sizes are sufficiently larger than the stable rank of $\mathbf{K}^{\frac{1}{2}}$, the error starts decaying at an exponential rate. Note that Theorem 1 only guarantees accuracy of our expressions for sketch sizes below the stable rank, however the predictions are accurate regardless of this constraint.

**Varying sketch type.**  In the next set of empirical results, we compare the performance of Gaussian and Rademacher sketches, and also verify the theory when sketching the data matrix $\mathbf{A}$ without kernel expansion, plotting $\|\mathbf{A} - \mathbf{A}(\mathbf{SA})^{\dagger}\mathbf{SA}\|_F^2 / \|\mathbf{A}\|_F^2$. Since both of the sketching methods have sub-gaussian entries, Corollary 1 predicts that they should have comparable performance in this task and match our expressions. This is exactly what we observe in Figure 3 for two datasets and a range of sketching sizes, as well as in other empirical results shown in Appendix C.

## 5   Conclusions

We derived the first theoretically supported precise expressions for the expected residual projection matrix, which is a central component in the analysis of RandNLA dimensionality reduction via sketching. Our analysis provides a new understanding of low-rank approximation, the Nyström method, and the convergence properties of many randomized iterative algorithms. As a direction for future work, we conjecture that our main result can be extended to sketch sizes larger than the stable rank of the data matrix.

## Broader Impact

In this paper, we investigate the spectral properties of residual (random) projection matrices, commonly appearing in various sketching-based methods. The precise theoretical description given in this paper provides performance guarantees for popular algorithms such as low-rank approximation and many randomized (iterative) optimization methods, and contributes to the development of more robust and reliable large-scale learning systems. The theoretical framework developed in this work presents no foreseeable negative societal consequence.

**Acknowledgments.**  We would like to acknowledge DARPA, NSF, and ONR for providing partial support of this work.

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
