[Supplementary Material]

# A  Proof of Theorem 2

We first introduce the following technical lemmas.

**Lemma 1.** *For $\mathbf{X} \in \mathbb{R}^{k \times n}$ with $k < n$, denote $\mathbf{P} = \mathbf{X}^\dagger \mathbf{X}$ and $\mathbf{P}_{-k} = \mathbf{X}_{-k}^\dagger \mathbf{X}_{-k}$, with $\mathbf{X}_{-i} \in \mathbb{R}^{(k-1) \times n}$ the matrix $\mathbf{X}$ without its $i$-th row $\mathbf{x}_i \in \mathbb{R}^n$. Then, conditioned on the event $E_k : \left\{ \left| \frac{\operatorname{tr}\boldsymbol{\Sigma}(\mathbf{I} - \mathbf{P}_{-k})}{\mathbf{x}_k^\top (\mathbf{I} - \mathbf{P}_{-k})\mathbf{x}_k} - 1 \right| \leq \frac{1}{2} \right\}$:*

$$(\mathbf{X}^\top \mathbf{X})^\dagger \mathbf{x}_k = \frac{(\mathbf{I} - \mathbf{P}_{-k})\mathbf{x}_k}{\mathbf{x}_k^\top (\mathbf{I} - \mathbf{P}_{-k})\mathbf{x}_k}, \quad \mathbf{P} - \mathbf{P}_{-k} = \frac{(\mathbf{I} - \mathbf{P}_{-k})\mathbf{x}_k \mathbf{x}_k^\top (\mathbf{I} - \mathbf{P}_{-k})}{\mathbf{x}_k^\top (\mathbf{I} - \mathbf{P}_{-k})\mathbf{x}_k}.$$

*Proof.* Since conditioned on $E_k$ we have $\mathbf{x}_k^\top (\mathbf{I} - \mathbf{P}_{-k})\mathbf{x}_k \neq 0$, from [Mey73, Theorem 1] we deduce

$$(\mathbf{X}^\top \mathbf{X})^\dagger = (\mathbf{A} + \mathbf{x}_k \mathbf{x}_k^\top)^\dagger = \mathbf{A}^\dagger - \frac{\mathbf{A}^\dagger \mathbf{x}_k \mathbf{x}_k^\top (\mathbf{I} - \mathbf{P}_{-k})}{\mathbf{x}_k^\top (\mathbf{I} - \mathbf{P}_{-k})\mathbf{x}_k} - \frac{(\mathbf{I} - \mathbf{P}_{-k})\mathbf{x}_k \mathbf{x}_k^\top \mathbf{A}^\dagger}{\mathbf{x}_k^\top (\mathbf{I} - \mathbf{P}_{-k})\mathbf{x}_k}$$
$$+ (1 + \mathbf{x}_k^\top \mathbf{A}^\dagger \mathbf{x}_k) \frac{(\mathbf{I} - \mathbf{P}_{-k})\mathbf{x}_k \mathbf{x}_k^\top (\mathbf{I} - \mathbf{P}_{-k})}{(\mathbf{x}_k^\top (\mathbf{I} - \mathbf{P}_{-k})\mathbf{x}_k)^2}$$

for $\mathbf{A} = \mathbf{X}_{-k}^\top \mathbf{X}_{-k}$ so that $\mathbf{I} - \mathbf{P}_{-k} = \mathbf{I} - \mathbf{A}^\dagger \mathbf{A}$, where we used the fact that $\mathbf{I} - \mathbf{P}_{-k}$ is a projection matrix so that $(\mathbf{I} - \mathbf{P}_{-k})^2 = \mathbf{I} - \mathbf{P}_{-k}$. As a consequence, multiplying by $\mathbf{x}_k$ and simplifying we get

$$(\mathbf{X}^\top \mathbf{X})^\dagger \mathbf{x}_k = \frac{(\mathbf{I} - \mathbf{P}_{-k})\mathbf{x}_k}{\mathbf{x}_k^\top (\mathbf{I} - \mathbf{P}_{-k})\mathbf{x}_k}.$$

By definition of the pseudoinverse, $\mathbf{P} = \mathbf{X}^\dagger \mathbf{X} = (\mathbf{X}^\top \mathbf{X})^\dagger \mathbf{X}^\top \mathbf{X}$ so that

$$\mathbf{P} - \mathbf{P}_{-k} = \mathbf{X}^\dagger \mathbf{X} - \mathbf{X}_{-k}^\dagger \mathbf{X}_{-k} = \frac{(\mathbf{I} - \mathbf{P}_{-k})\mathbf{x}_k \mathbf{x}_k^\top (\mathbf{I} - \mathbf{P}_{-k})}{\mathbf{x}_k^\top (\mathbf{I} - \mathbf{P}_{-k})\mathbf{x}_k}$$

where we used $\mathbf{A}(\mathbf{I} - \mathbf{P}_{-k}) = \mathbf{A} - \mathbf{A}\mathbf{A}^\dagger \mathbf{A} = 0$ and thus the conclusion. $\square$

**Lemma 2.** *For a $K$-sub-gaussian random vector $\mathbf{x} \in \mathbb{R}^n$ with $\mathbb{E}[\mathbf{x}] = 0$, $\mathbb{E}[\mathbf{x}\mathbf{x}^\top] = \mathbf{I}_n$ and positive semi-definite matrix $\mathbf{A} \in \mathbb{R}^{n \times n}$, we have*

$$\Pr\left[ |\mathbf{x}^\top \mathbf{A} \mathbf{x} - \operatorname{tr}\mathbf{A}| \geq \frac{1}{3}\operatorname{tr}\mathbf{A} \right] \leq 2\exp\left( -\min\left\{ \frac{r_{\mathbf{A}}}{9C^2 K^4}, \frac{\sqrt{r_{\mathbf{A}}}}{3CK^2} \right\} \right)$$

*with $r_{\mathbf{A}} = \operatorname{tr}\mathbf{A}/\|\mathbf{A}\|$ the stable rank of $\mathbf{A}$, and*

$$\mathbb{E}\left[ (\mathbf{x}^\top \mathbf{A} \mathbf{x} - \operatorname{tr}\mathbf{A})^2 \right] \leq c\, K^4\, \operatorname{tr}\mathbf{A}^2$$

*for some $C, c > 0$ independent of $K$.*

*Proof.* This follows from a Hanson-Wright type [RV13] sub-gaussian concentration inequality. More precisely, from [Zaj18, Corollary 2.9] we have, for $K$-sub-gaussian $\mathbf{x} \in \mathbb{R}^n$ with $\mathbb{E}[\mathbf{x}] = 0$, $\mathbb{E}[\mathbf{x}\mathbf{x}^\top] = \mathbf{I}_n$ and symmetric positive semi-definite $\mathbf{A} \in \mathbb{R}^{n \times n}$ that

$$\Pr\left\{ |\mathbf{x}^\top \mathbf{A} \mathbf{x} - \operatorname{tr}\mathbf{A}| \geq t \right\} \leq 2\exp\left( -\min\left\{ \frac{t^2}{C^2 K^4 \operatorname{tr}\mathbf{A}^2}, \frac{t}{CK^2 \sqrt{\operatorname{tr}\mathbf{A}^2}} \right\} \right)$$

for some universal constant $C > 0$. Taking $t = \frac{1}{3}\operatorname{tr}\mathbf{A}$ we have

$$\frac{t^2}{C^2 K^4 \operatorname{tr}\mathbf{A}^2} = \frac{(\operatorname{tr}\mathbf{A})^2}{9C^2 K^4 \operatorname{tr}\mathbf{A}^2} \geq \frac{\operatorname{tr}\mathbf{A}}{9C^2 K^4 \|\mathbf{A}\|} = \frac{r_{\mathbf{A}}}{9C^2 K^4}, \quad \frac{t}{CK^2 \sqrt{\operatorname{tr}\mathbf{A}^2}} \geq \frac{\sqrt{r_{\mathbf{A}}}}{3CK^2}$$

where we use the fact that $\operatorname{tr}\mathbf{A}^2 \leq \|\mathbf{A}\|\operatorname{tr}\mathbf{A}$.

Integrating this bound yields:

$$\mathbb{E}\left[ (\mathbf{x}^\top \mathbf{A} \mathbf{x} - \operatorname{tr}\mathbf{A})^2 \right] \leq c\, K^4\, \operatorname{tr}\mathbf{A}^2$$

and thus the conclusion. $\square$

**Lemma 3.** *With the notations of Lemma 1, for $X = \operatorname{tr} \mathbf{\Sigma}(\mathbf{P}_{-k} - \mathbb{E}[\mathbf{P}_{-k}])$ and $\|\mathbf{\Sigma}\| = 1$, we have*

$$\mathbb{E}[X^2] \leq Ck \quad \text{and} \quad \Pr\{|X| \geq t\} \leq 2\mathrm{e}^{-\frac{t^2}{ck}}.$$

*for some universal constant $C, c > 0$.*

*Proof.* To simplify notations, we work on $\mathbf{P}$ instead of $\mathbf{P}_{-k}$, the same line of argument applies to $\mathbf{P}_{-k}$ by changing the sample size $k$ to $k - 1$.

First note that

$$X = \operatorname{tr}\mathbf{\Sigma}(\mathbf{P} - \mathbb{E}\mathbf{P}) = \mathbb{E}_k[\operatorname{tr}\mathbf{\Sigma}\mathbf{P}] - \mathbb{E}_0[\operatorname{tr}\mathbf{\Sigma}\mathbf{P}]$$
$$= \sum_{i=1}^k \left(\mathbb{E}_i[\operatorname{tr}\mathbf{\Sigma}\mathbf{P}] - \mathbb{E}_{i-1}[\operatorname{tr}\mathbf{\Sigma}\mathbf{P}]\right) = \sum_{i=1}^k (\mathbb{E}_i - \mathbb{E}_{i-1})\operatorname{tr}\mathbf{\Sigma}(\mathbf{P} - \mathbf{P}_{-i})$$

where we used the fact that $\mathbb{E}_i[\operatorname{tr}\mathbf{\Sigma}\mathbf{P}_{-i}] = \mathbb{E}_{i-1}[\operatorname{tr}\mathbf{\Sigma}\mathbf{P}_{-i}]$, for $\mathbb{E}_i[\cdot]$ the conditional expectation with respect to $\mathcal{F}_i$ the $\sigma$-field generating the rows $\mathbf{x}_1 \ldots, \mathbf{x}_i$ of $\mathbf{X}$. This forms a martingale difference sequence (it is a difference sequence of the Doob martingale for $\operatorname{tr}\mathbf{\Sigma}(\mathbf{P} - \mathbf{P}_{-i})$ with respect to filtration $\mathcal{F}_i$) hence it falls within the scope of the Burkholder inequality [Bur73], recalled as follows.

**Lemma 4.** *For $\{x_i\}_{i=1}^k$ a real martingale difference sequence with respect to the increasing $\sigma$ field $\mathcal{F}_i$, we have, for $L > 1$, there exists $C_L > 0$ such that*

$$\mathbb{E}\left[\left|\sum_{i=1}^k x_i\right|^L\right] \leq C_L\mathbb{E}\left[\left(\sum_{i=1}^k |x_i|^2\right)^{L/2}\right].$$

From Lemma 1, $\mathbf{P} - \mathbf{P}_{-i} = \frac{(\mathbf{I}-\mathbf{P}_{-i})\mathbf{x}_i\mathbf{x}_i^\top(\mathbf{I}-\mathbf{P}_{-i})}{\mathbf{x}_i^\top(\mathbf{I}-\mathbf{P}_{-i})\mathbf{x}_i}$ is positive semi-definite, we have $\operatorname{tr}\mathbf{\Sigma}(\mathbf{P}-\mathbf{P}_{-i}) \leq \|\mathbf{\Sigma}\| = 1$ so that with Lemma 4 we obtain with $x_i = (\mathbb{E}_i - \mathbb{E}_{i-1})\operatorname{tr}\mathbf{\Sigma}(\mathbf{P} - \mathbf{P}_{-i})$ that, for $L > 1$

$$\mathbb{E}|X|^L \leq C_L k^{L/2}.$$

In particular, for $L = 2$, we obtain $\mathbb{E}|X|^2 \leq Ck$.

For the second result, since we have almost surely bounded martingale differences ($|x_i| \leq 2$), by the Azuma-Hoeffding inequality

$$\Pr\{|X| \geq t\} \leq 2\mathrm{e}^{\frac{-t^2}{8k}}$$

as desired. $\qquad\qquad\square$

## A.1 Complete proof of Theorem 2

Equipped with the lemmas above, we are ready to prove Theorem 2. First note that:

1. Since $\mathbf{X}^\dagger\mathbf{X} \overset{d}{=} (\alpha\mathbf{X})^\dagger(\alpha\mathbf{X})$ for any $\alpha \in \mathbb{R} \setminus \{0\}$, we can assume without loss of generality (after rescaling $\bar{\mathbf{P}}_\perp$ correspondingly) that $\|\mathbf{\Sigma}\| = 1$.

2. According to the definition of $\bar{\mathbf{P}}_\perp$ and $\gamma$, the following bounds hold

$$\frac{1}{\gamma + 1}\mathbf{I} \preceq \bar{\mathbf{P}}_\perp \preceq \mathbf{I}, \quad \gamma \leq \frac{k}{r - k} = \frac{1}{\rho - 1} \tag{7}$$

for $r \equiv \frac{\operatorname{tr}\mathbf{\Sigma}}{\|\mathbf{\Sigma}\|} = \operatorname{tr}\mathbf{\Sigma}$ and $\rho \equiv \frac{r}{k} > 1$, where we used the fact that

$$k = n - \operatorname{tr}\bar{\mathbf{P}}_\perp = \operatorname{tr}\bar{\mathbf{P}}_\perp(\gamma\mathbf{\Sigma} + \mathbf{I}) - \operatorname{tr}\bar{\mathbf{P}}_\perp = \gamma\operatorname{tr}\bar{\mathbf{P}}_\perp\mathbf{\Sigma} \geq \frac{\gamma}{\gamma + 1}\operatorname{tr}\mathbf{\Sigma},$$

so that $r = \operatorname{tr}\mathbf{\Sigma} \leq k \cdot \frac{\gamma + 1}{\gamma}$.

3. As already discussed in Section 2.1, to obtain the lower and upper bound for $\mathbb{E}[\mathbf{P}_\perp]$ in the sense of symmetric matrix as in Theorem 2, it suffices to bound the following spectral norm

$$\|\mathbf{I} - \mathbb{E}[\mathbf{P}_\perp]\bar{\mathbf{P}}_\perp^{-1}\| \leq \frac{C_\rho}{\sqrt{r}}, \tag{8}$$

so that, with $\frac{\rho-1}{\rho}\mathbf{I} \preceq \bar{\mathbf{P}}_\perp \preceq \mathbf{I}$ from (7), we have

$$\|\mathbf{I} - \bar{\mathbf{P}}_\perp^{-\frac{1}{2}}\mathbb{E}[\mathbf{P}_\perp]\bar{\mathbf{P}}_\perp^{-\frac{1}{2}}\| = \|\bar{\mathbf{P}}_\perp^{-\frac{1}{2}}(\mathbf{I} - \mathbb{E}[\mathbf{P}_\perp]\bar{\mathbf{P}}_\perp^{-1})\bar{\mathbf{P}}_\perp^{\frac{1}{2}}\| \leq \frac{C_\rho}{\sqrt{r}}\sqrt{\frac{\rho}{\rho-1}}.$$

Defining $\epsilon = \frac{C_\rho}{\sqrt{r}}\sqrt{\frac{\rho}{\rho-1}}$, this means that all eigenvalues of the p.s.d. matrix $\bar{\mathbf{P}}_\perp^{-\frac{1}{2}}\mathbb{E}[\mathbf{P}_\perp]\bar{\mathbf{P}}_\perp^{-\frac{1}{2}}$ lie in the interval $[1-\epsilon, 1+\epsilon]$, and

$$(1-\epsilon)\mathbf{I} \preceq \bar{\mathbf{P}}_\perp^{-\frac{1}{2}}\mathbb{E}[\mathbf{P}_\perp]\bar{\mathbf{P}}_\perp^{-\frac{1}{2}} \preceq (1+\epsilon)\mathbf{I}.$$

so that by multiplying $\bar{\mathbf{P}}_\perp^{\frac{1}{2}}$ on both sides, we obtain the desired bound.

As a consequence of the above observations, we only need to prove (8) under the setting $\|\boldsymbol{\Sigma}\| = 1$. The proof comes in the following two steps:

1. For $\mathbf{P}_{-i} = \mathbf{X}_{-i}^\dagger\mathbf{X}_{-i}$, with $\mathbf{X}_{-i} \in \mathbb{R}^{(k-1)\times n}$ the matrix $\mathbf{X}$ without its $i$-th row, we define, for $i \in \{1, \ldots, k\}$, the following events

$$E_i : \left\{ \left| \frac{\mathrm{tr}(\mathbf{I} - \mathbf{P}_{-i})\boldsymbol{\Sigma}}{\mathbf{x}_i^\top(\mathbf{I} - \mathbf{P}_{-i})\mathbf{x}_i} - 1 \right| \leq \frac{1}{2} \right\}, \tag{9}$$

   where we recall $\mathbf{x}_i \in \mathbb{R}^n$ is the $i$-th row of $\mathbf{X}$ so that $\mathbb{E}[\mathbf{x}_i] = 0$ and $\mathbb{E}[\mathbf{x}_i\mathbf{x}_i^\top] = \boldsymbol{\Sigma}$. With Lemma 2, we can bound the probability of $\neg E_i$, and consequently that of $\neg E$ for $E = \bigwedge_{i=1}^k E_i$;

2. We then bound, conditioned on $E$ and $\neg E$ respectively, the spectral norm $\|\mathbf{I} - \mathbb{E}[\mathbf{P}_\perp]\bar{\mathbf{P}}_\perp^{-1}\|$. More precisely, since

$$\begin{aligned}
\mathbf{I} - \mathbb{E}[\mathbf{P}_\perp]\bar{\mathbf{P}}_\perp^{-1} &= \mathbb{E}[\mathbf{P}] - \gamma\mathbb{E}[\mathbf{P}_\perp]\boldsymbol{\Sigma} \\
&= \mathbb{E}[\mathbf{P} \cdot \mathbf{1}_E] + \mathbb{E}[\mathbf{P} \cdot \mathbf{1}_{\neg E}] - \gamma\mathbb{E}[\mathbf{P}_\perp]\boldsymbol{\Sigma} \\
&= k\,\mathbb{E}\left[ \frac{(\mathbf{I} - \mathbf{P}_{-k})\mathbf{x}_k\mathbf{x}_k^\top}{\mathbf{x}_k^\top(\mathbf{I} - \mathbf{P}_{-k})\mathbf{x}_k} \cdot \mathbf{1}_E \right] - \gamma\mathbb{E}[\mathbf{P}_\perp]\boldsymbol{\Sigma} + \mathbb{E}[\mathbf{P} \cdot \mathbf{1}_{\neg E}] \\
&= \gamma\underbrace{\mathbb{E}\left[ (\bar{s} - \hat{s}) \cdot \frac{(\mathbf{I} - \mathbf{P}_{-k})\mathbf{x}_k\mathbf{x}_k^\top}{\mathbf{x}_k^\top(\mathbf{I} - \mathbf{P}_{-k})\mathbf{x}_k} \cdot \mathbf{1}_E \right]}_{\mathbf{T}_1} -\gamma\underbrace{\mathbb{E}[(\mathbf{I} - \mathbf{P}_{-k})\mathbf{x}_k\mathbf{x}_k^\top \cdot \mathbf{1}_{\neg E}]}_{\mathbf{T}_2} \\
&\quad + \gamma\underbrace{\mathbb{E}[\mathbf{P} - \mathbf{P}_{-k}]\boldsymbol{\Sigma}}_{\mathbf{T}_3} + \underbrace{\mathbb{E}[\mathbf{P} \cdot \mathbf{1}_{\neg E}]}_{\mathbf{T}_4},
\end{aligned}$$

   where we used Lemma 1 for the third equality and denote $\hat{s} = \mathbf{x}_k^\top(\mathbf{I} - \mathbf{P}_{-k})\mathbf{x}_k$ as well as $\bar{s} = \mathrm{tr}\bar{\mathbf{P}}_\perp\boldsymbol{\Sigma} = k/\gamma$. It then remains to bound the spectral norms of $\mathbf{T}_1, \mathbf{T}_2, \mathbf{T}_3, \mathbf{T}_4$ to reach the conclusion.

Another important relation that will be constantly used throughout the proof is

$$\mathrm{tr}(\mathbf{I} - \mathbf{P}_{-k})\boldsymbol{\Sigma} = \mathrm{tr}\boldsymbol{\Sigma}^{\frac{1}{2}}(\mathbf{I} - \mathbf{P}_{-k})^2\boldsymbol{\Sigma}^{\frac{1}{2}} = \|\boldsymbol{\Sigma}^{\frac{1}{2}} - \boldsymbol{\Sigma}^{\frac{1}{2}}\mathbf{X}_{-k}^\dagger\mathbf{X}_{-k}\|_F^2 \geq \sum_{i\geq k}\lambda_i(\boldsymbol{\Sigma}) \geq r - k \tag{10}$$

where we used the fact that $\mathrm{rank}(\mathbf{X}_{-k}^\dagger\mathbf{X}_{-k}) \leq \mathrm{rank}(\mathbf{X}_{-k}) \leq k - 1$ and arranged the eigenvalues $1 = \lambda_1(\boldsymbol{\Sigma}) \geq \ldots \geq \lambda_n(\boldsymbol{\Sigma})$ in a non-increasing order. As a consequence, we also have

$$\frac{\mathrm{tr}(\mathbf{I} - \mathbf{P}_{-k})\boldsymbol{\Sigma}}{\|(\mathbf{I} - \mathbf{P}_{-k})\boldsymbol{\Sigma}\|} \geq \mathrm{tr}(\mathbf{I} - \mathbf{P}_{-k})\boldsymbol{\Sigma} \geq r - k. \tag{11}$$

For the first step, we have, with Lemma 2 and (11) that

$$\Pr(\neg E_i) \le \Pr\left\{ |\mathbf{x}_i^\top(\mathbf{I}-\mathbf{P}_{-i})\mathbf{x}_i - \mathrm{tr}\mathbf{\Sigma}(\mathbf{I}-\mathbf{P}_{-i})| \ge \frac{1}{3}\mathrm{tr}\mathbf{\Sigma}(\mathbf{I}-\mathbf{P}_{-i}) \right\}$$

$$\le 2\mathrm{e}^{-\min\left\{ \frac{r-k}{9C^2K^4}, \frac{\sqrt{r-k}}{3CK^2} \right\}}.$$

so that with the union bound we obtain

$$\Pr(\neg E) \le 2k\mathrm{e}^{-\min\left\{ \frac{r-k}{9C^2K^4}, \frac{\sqrt{r-k}}{3CK^2} \right\}} \le \frac{k}{(r-k)^2} \cdot 2(r-k)^2 \mathrm{e}^{-\min\left\{ \frac{r-k}{9C^2K^4}, \frac{\sqrt{r-k}}{3CK^2} \right\}} \le \frac{C_\rho}{r-k} \quad (12)$$

where we used the fact that, for $\alpha > 0$, $x^2\mathrm{e}^{-\alpha x} \le \frac{4\mathrm{e}^{-2}}{\alpha^2}$ and $x^4\mathrm{e}^{-\alpha x} \le \frac{256\mathrm{e}^{-4}}{\alpha^4}$ on $x > 0$. Also, denote $c_\rho = \frac{r-k}{r} = \frac{\rho-1}{\rho} > 0$, we have

$$\Pr(\neg E) \le \frac{C_\rho}{r-k} = \frac{C_\rho}{c_\rho r} = \frac{C_\rho'}{r} \quad (13)$$

for some $C_\rho' > 0$ that depends on $\rho = r/k > 1$ and the sub-gaussian norm $K$.

At this point, note that, conditioned on the event $E$, we have for $i \in \{1, \dots, k\}$

$$\frac{1}{2}\frac{1}{\mathrm{tr}(\mathbf{I}-\mathbf{P}_{-i})\mathbf{\Sigma}} \le \frac{1}{\mathbf{x}_i^\top(\mathbf{I}-\mathbf{P}_{-i})\mathbf{x}_i} \le \frac{3}{2}\frac{1}{\mathrm{tr}(\mathbf{I}-\mathbf{P}_{-i})\mathbf{\Sigma}}, \quad (14)$$

Also, with (13) and the fact that $\|\mathbf{P}\| \le 1$, we have $\|\mathbf{T}_4\| \le \frac{C_\rho}{r}$ for some $C_\rho > 0$ that depends on $\rho$ and $K$. To handle non-symmetric matrix $\mathbf{T}_2$, note that $\mathbf{T}_2 + \mathbf{T}_2^\top$ is symmetric and

$$-\mathbb{E}[(\mathbf{I}-\mathbf{P}_{-k})\cdot\mathbf{1}_{\neg E}]-\mathbb{E}[(\mathbf{x}_k^\top\mathbf{x}_k)\mathbf{x}_k\mathbf{x}_k^\top\cdot\mathbf{1}_{\neg E}] \preceq \mathbf{T}_2+\mathbf{T}_2^\top \preceq \mathbb{E}[(\mathbf{I}-\mathbf{P}_{-k})\cdot\mathbf{1}_{\neg E}]+\mathbb{E}[(\mathbf{x}_k^\top\mathbf{x}_k)\mathbf{x}_k\mathbf{x}_k^\top\cdot\mathbf{1}_{\neg E}]$$
$$(15)$$

with $-(\mathbf{AA}^\top + \mathbf{BB}^\top) \preceq \mathbf{AB}^\top + \mathbf{BA}^\top \preceq \mathbf{AA}^\top + \mathbf{BB}^\top$. To obtain an upper bound for operator norm of $\mathbb{E}[(\mathbf{x}_k^\top\mathbf{x}_k)\mathbf{x}_k\mathbf{x}_k^\top \cdot \mathbf{1}_{\neg E}]$, note that

$$\|\mathbb{E}[(\mathbf{x}_k^\top\mathbf{x}_k)\mathbf{x}_k\mathbf{x}_k^\top \cdot \mathbf{1}_{\neg E}]\| \le \mathbb{E}[(\mathbf{x}_k^\top\mathbf{x}_k)^2 \cdot \mathbf{1}_{\neg E}] = \int_0^\infty \Pr(\mathbf{x}^\top\mathbf{x} \cdot \mathbf{1}_{\neg E} \ge \sqrt{t})dt$$

$$\le \int_0^X \Pr(\mathbf{x}^\top\mathbf{x} \cdot \mathbf{1}_{\neg E} \ge \sqrt{t})dt + \int_X^\infty \Pr(\mathbf{x}^\top\mathbf{x} \ge \sqrt{t})dt$$

$$\le X \cdot \Pr(\neg E) + \int_X^\infty \mathrm{e}^{-\min\left\{ \frac{t}{C^2K^4r}, \frac{\sqrt{t}}{CK^2\sqrt{r}} \right\}}dt \le \frac{c_\rho}{r}$$

where we recall $\mathbb{E}[\mathbf{x}^\top\mathbf{x}] = \mathrm{tr}\mathbf{\Sigma} = r$ and take $X \ge C^2K^4r$, the third line follows from the proof of Lemma 2 and the forth line from the same argument as in (12). Moreover, since $\|\mathbf{T}_2\| \le \|\mathbf{T}_2 + \mathbf{T}_2^\top\|$ (see for example [Ser10, Proposition 5.11]), we conclude that $\|\mathbf{T}_2\| \le \frac{C_\rho}{r}$.

And it thus remains to handle the terms $\mathbf{T}_1$ and $\mathbf{T}_3$ to obtain a bound on $\|\mathbf{I} - \mathbb{E}[\mathbf{P}_\perp]\bar{\mathbf{P}}_\perp^{-1}\|$.

To bound $\mathbf{T}_3$, with $\mathbf{P} - \mathbf{P}_{-k} = \frac{(\mathbf{I}-\mathbf{P}_{-k})\mathbf{x}_k\mathbf{x}_k^\top(\mathbf{I}-\mathbf{P}_{-k})}{\mathbf{x}_k^\top(\mathbf{I}-\mathbf{P}_{-k})\mathbf{x}_k}$ in Lemma 1, we have

$$\|\mathbf{T}_3\| \le \left\| \mathbb{E}\left[ \frac{(\mathbf{I}-\mathbf{P}_{-k})\mathbf{x}_k\mathbf{x}_k^\top(\mathbf{I}-\mathbf{P}_{-k})}{\mathbf{x}_k^\top(\mathbf{I}-\mathbf{P}_{-k})\mathbf{x}_k} \cdot \mathbf{1}_E \right] \right\| + \|\mathbb{E}[(\mathbf{P}-\mathbf{P}_{-k})\cdot\mathbf{1}_{\neg E}]\|$$

$$\le \frac{3}{2}\mathbb{E}\left[ \frac{1}{\mathrm{tr}(\mathbf{I}-\mathbf{P}_{-k})\mathbf{\Sigma}} \right] + \frac{c_\rho}{r-k} \le \frac{C_\rho}{r-k} = \frac{C_\rho'}{r}$$

where we used the fact that $\mathrm{tr}\,(\mathbf{I}-\mathbf{P}_{-k})\mathbf{\Sigma} \ge r-k$ from (10) and recall $\rho \equiv r/k > 1$.

For $\mathbf{T}_1$ we write

$$\|\mathbf{T}_1\| \le \mathbb{E}\left[ \|\mathbf{I}-\mathbf{P}_{-k}\| \cdot \left\| \mathbb{E}\left[ |\bar{s}-\hat{s}| \cdot \frac{\mathbf{x}_k\mathbf{x}_k^\top}{\mathbf{x}_k^\top(\mathbf{I}-\mathbf{P}_{-k})\mathbf{x}_k} \cdot \mathbf{1}_E \mid \mathbf{P}_{-k} \right] \right\| \right]$$

$$\le \frac{3}{2}\frac{1}{r-k} \cdot \mathbb{E}\left[ \sup_{\|\mathbf{v}\|=1} \mathbb{E}\left[ |\bar{s}-\hat{s}| \cdot \mathbf{v}^\top\mathbf{x}_k\mathbf{x}_k^\top\mathbf{v} \cdot \mathbf{1}_E \mid \mathbf{P}_{-k} \right] \right]$$

$$\le \frac{C_\rho}{r} \cdot \mathbb{E}\left[ \underbrace{\sqrt{\mathbb{E}[(\bar{s}-\hat{s})^2 \cdot \mathbf{1}_E \mid \mathbf{P}_{-k}]}}_{T_{1,1}} \cdot \underbrace{\sup_{\|\mathbf{v}\|=1} \sqrt{\mathbb{E}[(\mathbf{v}^\top\mathbf{x}_k)^4]}}_{T_{1,2}} \right]$$

where we used Jensen's inequality for the first inequality, the relation in (10) for the second inequality, and Cauchy–Schwarz for the third inequality.

We first bound $T_{1,2}$ by definition of sub-gaussian random vectors. We have for $\mathbf{x}_k$ a $K$-sub-gaussian and $\|\mathbf{v}\| = 1$ that, $\mathbf{v}^\top \mathbf{x}_k$ is a sub-gaussian random variable with $\|\mathbf{v}^\top \mathbf{a}\|_{\psi_2} \le K$. As such, $T_{1,2} \le CK^2$ for some absolute constant $C > 0$, see for example [Ver18, Section 2.5.2].

For $T_{1,1}$ we have

$$\sqrt{\mathbb{E}[(\bar{s} - \hat{s})^2 \cdot \mathbf{1}_E \mid \mathbf{P}_{-k}]} = \sqrt{(\bar{s} - s)^2 + \mathbb{E}[(s - \hat{s})^2 \cdot \mathbf{1}_E]}$$

where we denote $s = \mathbb{E}[\hat{s}] = \operatorname{tr} \mathbb{E}[\mathbf{I} - \mathbf{P}_{-k}]\boldsymbol{\Sigma}$. Note that

$$\mathbb{E}\big[(s - \hat{s})^2\big] = \mathbb{E}\big[\big(\operatorname{tr}\boldsymbol{\Sigma}(\mathbf{P}_{-k} - \mathbb{E}[\mathbf{P}_{-k}])\big)^2\big] + \mathbb{E}\big[(\operatorname{tr}(\mathbf{I} - \mathbf{P}_{-k})\boldsymbol{\Sigma} - \mathbf{x}_k^\top(\mathbf{I} - \mathbf{P}_{-k})\mathbf{x}_k)^2\big]$$
$$\le C_1 k + C_2 \mathbb{E}\big[\operatorname{tr}(\boldsymbol{\Sigma} - \mathbf{P}_{-k}\boldsymbol{\Sigma})^2\big] \le C(k + s) \le C\big(k + \bar{s} + |s - \bar{s}|\big)$$

where we used Lemma 3 and Lemma 2. Recall that $\bar{s} = \operatorname{tr}\bar{\mathbf{P}}_\perp \boldsymbol{\Sigma} \le \operatorname{tr}\boldsymbol{\Sigma} = r$ and $k < r$, we have

$$T_{1,1} \le \sqrt{(\bar{s} - s)^2 + C(|\bar{s} - s| + 2r)} \tag{16}$$

It remains to bound $|\bar{s} - s|$. Note that $\mathbf{P} = (\mathbf{X}^\top \mathbf{X})^\dagger \mathbf{X}^\top \mathbf{X} = \mathbf{X}^\top \mathbf{X}(\mathbf{X}^\top \mathbf{X})^\dagger$ and is symmetric, so

$$\mathbf{I} - \mathbb{E}[\mathbf{P}_\perp]\bar{\mathbf{P}}_\perp^{-1} + \mathbf{I} - \bar{\mathbf{P}}_\perp^{-1}\mathbb{E}[\mathbf{P}_\perp] = 2\mathbb{E}[\mathbf{P}] - \mathbb{E}[\gamma\mathbf{P}_\perp\boldsymbol{\Sigma}] - \mathbb{E}[\gamma\boldsymbol{\Sigma}\mathbf{P}_\perp]$$

$$= \sum_{i=1}^k \mathbb{E}\big[(\mathbf{X}^\top \mathbf{X})^\dagger \mathbf{x}_i \mathbf{x}_i^\top + \mathbf{x}_i \mathbf{x}_i^\top (\mathbf{X}^\top \mathbf{X})^\dagger\big] - \gamma(\mathbb{E}[\mathbf{P}_\perp]\boldsymbol{\Sigma} + \boldsymbol{\Sigma}\mathbb{E}[\mathbf{P}_\perp])$$

$$= \gamma \mathbb{E}\left[\bar{s} \cdot \frac{(\mathbf{I} - \mathbf{P}_{-k})\mathbf{x}_k \mathbf{x}_k^\top + \mathbf{x}_k \mathbf{x}_k^\top(\mathbf{I} - \mathbf{P}_{-k})}{\mathbf{x}_k^\top(\mathbf{I} - \mathbf{P}_{-k})\mathbf{x}_k}\right] - \gamma \mathbb{E}\left[\hat{s} \cdot \frac{(\mathbf{I} - \mathbf{P}_{-k})\mathbf{x}_k \mathbf{x}_k^\top + \mathbf{x}_k \mathbf{x}_k^\top(\mathbf{I} - \mathbf{P}_{-k})}{\mathbf{x}_k^\top(\mathbf{I} - \mathbf{P}_{-k})\mathbf{x}_k}\right]$$

$$+ \gamma\left(\mathbb{E}[(\mathbf{I} - \mathbf{P}_{-k})\boldsymbol{\Sigma}] + \mathbb{E}[\boldsymbol{\Sigma}(\mathbf{I} - \mathbf{P}_{-k})]\right) - \gamma(\mathbb{E}[\mathbf{P}_\perp]\boldsymbol{\Sigma} + \boldsymbol{\Sigma}\mathbb{E}[\mathbf{P}_\perp])$$

$$= \gamma \mathbb{E}\left[(\bar{s} - \hat{s}) \cdot \frac{(\mathbf{I} - \mathbf{P}_{-k})\mathbf{x}_k \mathbf{x}_k^\top + \mathbf{x}_k \mathbf{x}_k^\top(\mathbf{I} - \mathbf{P}_{-k})}{\mathbf{x}_k^\top(\mathbf{I} - \mathbf{P}_{-k})\mathbf{x}_k}\right] + \gamma(\mathbb{E}[\mathbf{P} - \mathbf{P}_{-k}]\boldsymbol{\Sigma} + \boldsymbol{\Sigma}\mathbb{E}[\mathbf{P} - \mathbf{P}_{-k}]).$$

Moreover, using the fact that $\bar{\mathbf{P}}_\perp \boldsymbol{\Sigma} \preceq \frac{1}{\gamma+1}\mathbf{I}$ and $\bar{\mathbf{P}}_\perp \boldsymbol{\Sigma} = \boldsymbol{\Sigma}\bar{\mathbf{P}}_\perp$, we obtain that

$$|\bar{s} - s| = |\operatorname{tr}(\bar{\mathbf{P}}_\perp - \mathbb{E}[\mathbf{I} - \mathbf{P}_{-k}])\boldsymbol{\Sigma}| \le |\operatorname{tr}(\bar{\mathbf{P}}_\perp - \mathbb{E}[\mathbf{P}_\perp])\boldsymbol{\Sigma}| + |\operatorname{tr}\mathbb{E}[\mathbf{P} - \mathbf{P}_{-k}]\boldsymbol{\Sigma}|$$

$$= \frac{1}{2}\left|\operatorname{tr}(\mathbf{I} - \mathbb{E}[\mathbf{P}_\perp]\bar{\mathbf{P}}_\perp^{-1})\bar{\mathbf{P}}_\perp \boldsymbol{\Sigma} + \operatorname{tr}\bar{\mathbf{P}}_\perp(\mathbf{I} - \bar{\mathbf{P}}_\perp^{-1}\mathbb{E}[\mathbf{P}_\perp])\boldsymbol{\Sigma}\right| + \operatorname{tr}\mathbb{E}\left[\frac{(\mathbf{I} - \mathbf{P}_{-k})\mathbf{x}_k \mathbf{x}_k^\top(\mathbf{I} - \mathbf{P}_{-k})}{\mathbf{x}_k^\top(\mathbf{I} - \mathbf{P}_{-k})\mathbf{x}_k}\right]\boldsymbol{\Sigma}$$

$$\le \frac{1}{2}\left|\operatorname{tr}(\mathbf{I} - \mathbb{E}[\mathbf{P}_\perp]\bar{\mathbf{P}}_\perp^{-1} + \mathbf{I} - \bar{\mathbf{P}}_\perp^{-1}\mathbb{E}[\mathbf{P}_\perp])\bar{\mathbf{P}}_\perp \boldsymbol{\Sigma}\right| + 1$$

$$\le \frac{\gamma}{2}\mathbb{E}\left[|\bar{s} - \hat{s}| \cdot \frac{\operatorname{tr}((\mathbf{I} - \mathbf{P}_{-k})\mathbf{x}_k \mathbf{x}_k^\top + \mathbf{x}_k \mathbf{x}_k^\top(\mathbf{I} - \mathbf{P}_{-k}))\bar{\mathbf{P}}_\perp \boldsymbol{\Sigma}}{\operatorname{tr}(\mathbf{I} - \mathbf{P}_{-k})\mathbf{x}_k \mathbf{x}_k^\top}\right]$$

$$+ \gamma \mathbb{E}\left[\frac{\operatorname{tr}(\mathbf{I} - \mathbf{P}_{-k})\mathbf{x}_k \mathbf{x}_k^\top(\mathbf{I} - \mathbf{P}_{-k})\bar{\mathbf{P}}_\perp \boldsymbol{\Sigma}}{\operatorname{tr}(\mathbf{I} - \mathbf{P}_{-k})\mathbf{x}_k \mathbf{x}_k^\top}\right] + 1$$

$$\le \frac{\gamma}{\gamma+1}\left(\mathbb{E}\left[|\bar{s} - \hat{s}| \cdot \frac{\mathbf{x}_k^\top(\mathbf{I} - \mathbf{P}_{-k})\mathbf{x}_k}{\mathbf{x}_k^\top(\mathbf{I} - \mathbf{P}_{-k})\mathbf{x}_k}\right] + 1\right) + 1 \le \frac{\gamma}{\gamma+1}\left(|\bar{s} - s| + \mathbb{E}\big[|s - \hat{s}|\big] + 1\right) + 1$$

$$\le \frac{\gamma}{\gamma+1}\left(|\bar{s} - s| + C\sqrt{|\bar{s} - s|} + C\sqrt{2r} + 1\right) + 1.$$

Solving for $|\bar{s} - s|$, we deduce that

$$|\bar{s} - s| \le C_1 \sqrt{r} + C_2,$$

so plugging back to (16) we get $T_{1,1} \le C\sqrt{r}$ and $\|\mathbf{T}_1\| \le \frac{C_\rho}{\sqrt{r}}$, thus completing the proof.

## B Convergence analysis of randomized iterative methods

Here, we discuss how our surrogate expressions for the expected residual projection can be used to perform convergence analysis for several randomized iterative optimization methods discussed in Section 1.3.

## B.1 Generalized Kaczmarz method

Generalized Kaczmarz [GR15] is an iterative method for solving an $m \times n$ linear system $\mathbf{A}\mathbf{x} = \mathbf{b}$, which uses a $k \times m$ sketching matrix $\mathbf{S}_t$ to reduce the linear system and update an iterate $\mathbf{x}^t$ as follows:

$$\mathbf{x}^{t+1} = \operatorname*{argmin}_{\mathbf{x}} \|\mathbf{x} - \mathbf{x}^t\|^2 \quad \text{subject to} \quad \mathbf{S}_t \mathbf{A} \mathbf{x} = \mathbf{S}_t \mathbf{b}.$$

Assume that $\mathbf{x}^*$ is the unique solution to the linear system $\mathbf{A}\mathbf{x} = \mathbf{b}$. In Theorems 4.1 and 4.6, [GR15] show that the expected trajectory of the generalized Kaczmarz iterates, as they converge to $\mathbf{x}^*$, is controlled by the projection matrix $\mathbf{P} = (\mathbf{S}_t \mathbf{A})^\dagger \mathbf{S}_t \mathbf{A}$ as follows:

([GR15], Theorem 4.1) $\qquad \mathbb{E}[\mathbf{x}^{t+1} - \mathbf{x}^*] = (\mathbf{I} - \mathbb{E}[\mathbf{P}]) \mathbb{E}[\mathbf{x}^t - \mathbf{x}^*],$

([GR15], Theorem 4.6) $\qquad \mathbb{E}\big[\|\mathbf{x}^{t+1} - \mathbf{x}^*\|^2\big] \le (1 - \kappa) \mathbb{E}\big[\|\mathbf{x}^t - \mathbf{x}^*\|^2\big],$ where $\kappa = \lambda_{\min}\big(\mathbb{E}[\mathbf{P}]\big).$

Both of these results depend on the expected projection $\mathbb{E}[\mathbf{P}]$. The first one describes the expected trajectory of the iterate, whereas the second one gives the worst-case convergence rate in terms of the so-called *stochastic condition number* $\kappa$. We next demonstrate how Theorem 1 can be used in combination with the above results to obtain convergence analysis for generalized Kaczmarz which is formulated in terms of the spectral properties of $\mathbf{A}$. This includes precise expressions for both the expected trajectory and $\kappa$. The following result is a more detailed version of Corollary 2 from Section 1.3.

**Corollary 3.** *Let $\sigma_i$ denote the singular values of $\mathbf{A}$, and let $k$ denote the size of sketch $\mathbf{S}_t$. Define:*

$$\Delta_t = \mathbf{x}^t - \mathbf{x}^* \quad and \quad \bar{\Delta}_{t+1} = (\gamma \mathbf{A}^\top \mathbf{A} + \mathbf{I})^{-1} \mathbb{E}[\Delta_t] \quad s.t. \quad \sum_i \frac{\gamma \sigma_i^2}{\gamma \sigma_i^2 + 1} = k.$$

*Suppose that $\mathbf{S}_t$ has i.i.d. mean-zero sub-gaussian entries and let $r = \|\mathbf{A}\|_F^2 / \|\mathbf{A}\|^2$ be the stable rank of $\mathbf{A}$. Assume that $\rho = r/k$ is a constant larger than 1. Then, the expected trajectory satisfies:*

$$\big\|\mathbb{E}[\Delta_{t+1}] - \bar{\Delta}_{t+1}\big\| \le \epsilon \cdot \|\bar{\Delta}_{t+1}\|, \quad for \quad \epsilon = O\big(\tfrac{1}{\sqrt{r}}\big). \tag{17}$$

*Moreover, we obtain the following worst-case convergence guarantee:*

$$\mathbb{E}\big[\|\Delta_{t+1}\|^2\big] \le \big(1 - (\bar{\kappa} - \epsilon)\big) \mathbb{E}\big[\|\Delta_t\|^2\big], \quad where \quad \bar{\kappa} = \frac{\sigma_{\min}^2}{\sigma_{\min}^2 + 1/\gamma}. \tag{18}$$

**Remark 2.** *Our worst-case convergence guarantee (18) requires the matrix $\mathbf{A}$ to be sufficiently well-conditioned so that $\bar{\kappa} - \epsilon > 0$. However, we believe that our surrogate expression $\bar{\kappa}$ for the stochastic condition number is far more accurate than suggested by the current analysis.*

## B.2 Randomized Subspace Newton

Randomized Subspace Newton (RSN, [GKLR19]) is a randomized Newton-type method for minimizing a smooth, convex and twice differentiable function $f : \mathbb{R}^d \times \mathbb{R}$. The iterative update for this algorithm is defined as follows:

$$\mathbf{x}^{t+1} = \mathbf{x}^t - \frac{1}{L} \mathbf{S}_t^\top (\mathbf{S}_t \mathbf{H}(\mathbf{x}^t) \mathbf{S}_t^\top)^\dagger \mathbf{S}_t \mathbf{g}(\mathbf{x}^t),$$

where $\mathbf{H}(\mathbf{x}^t)$ and $\mathbf{g}(\mathbf{x}^t)$ are the Hessian and gradient of $f$ at $\mathbf{x}^t$, respectively, whereas $\mathbf{S}_t$ is a $k \times d$ sketching matrix (with $k \ll d$) which is refreshed at every iteration. Here, $L$ denotes the *relative smoothness* constant defined by [GKLR19] in Assumption 1, which also defines relative strong convexity, denoted by $\mu$. In Theorem 2, they prove the following convergence guarantee for RSN:

$$\mathbb{E}[f(\mathbf{x}^t)] - f(\mathbf{x}^*) \le \Big(1 - \kappa \frac{\mu}{L}\Big)^t (f(\mathbf{x}^0) - f(\mathbf{x}^*)),$$

where $\kappa = \min_{\mathbf{x}} \kappa(\mathbf{x})$ and $\kappa(\mathbf{x}) = \lambda_{\min}^+(\mathbb{E}[\mathbf{P}(\mathbf{x})])$ is the smallest positive eigenvalue of the expectation of the projection matrix $\mathbf{P}(\mathbf{x}) = \mathbf{H}^{\frac{1}{2}}(\mathbf{x}) \mathbf{S}_t^\top (\mathbf{S}_t \mathbf{H}(\mathbf{x}) \mathbf{S}_t^\top)^\dagger \mathbf{S}_t \mathbf{H}^{\frac{1}{2}}(\mathbf{x})$. Our results lead to the following surrogate expression for this expected projection when the sketch is sub-gaussian:

$$\mathbb{E}[\mathbf{P}(\mathbf{x})] \simeq \mathbf{H}(\mathbf{x})\big(\mathbf{H}(\mathbf{x}) + \tfrac{1}{\gamma(\mathbf{x})}\mathbf{I}\big)^{-1} \quad \text{for} \quad \gamma(\mathbf{x}) > 0 \quad \text{s.t.} \quad \operatorname{tr}\mathbf{H}(\mathbf{x})\big(\mathbf{H}(\mathbf{x}) + \tfrac{1}{\gamma(\mathbf{x})}\mathbf{I}\big)^{-1} = k.$$

Thus, the condition number $\kappa$ of RSN can be estimated using the following surrogate expression:

$$\kappa \simeq \bar{\kappa} := \min_{\mathbf{x}} \frac{\lambda_{\min}^+(\mathbf{H}(\mathbf{x}))}{\lambda_{\min}^+(\mathbf{H}(\mathbf{x})) + 1/\gamma(\mathbf{x})}.$$

Just as in Corollary 3, an approximation of the form $|\bar{\kappa} - \kappa| \leq \epsilon$ can be shown from Theorem 1.

**Corollary 4.** *Suppose that sketch $\mathbf{S}_t$ has size $k$ and i.i.d. mean-zero sub-gaussian entries. Let $r = \min_{\mathbf{x}} \operatorname{tr} \mathbf{H}(\mathbf{x})/\|\mathbf{H}(\mathbf{x})\|$ be the (minimum) stable rank of the (square root) Hessian and assume that $\rho = r/k$ is a constant larger than 1. Then,*

$$|\kappa - \bar{\kappa}| \leq O\left(\tfrac{1}{\sqrt{r}}\right).$$

### B.3   Jacobian Sketching

Jacobian Sketching (JacSketch, [GRB20]) defines an $n \times n$ positive semi-definite weight matrix $\mathbf{W}$, and combines it with an $k \times n$ sketching matrix $\mathbf{S}$ (which is refreshed at every iteration of the algorithm), to implicitly construct the following projection matrix:

$$\Pi_{\mathbf{S}} = \mathbf{S}^\top (\mathbf{S}\mathbf{W}\mathbf{S}^\top)^\dagger \mathbf{S}\mathbf{W},$$

which is used to sketch the Jacobian at the current iterate (for the complete method, we refer to their Algorithm 1). The convergence rate guarantee given in their Theorem 3.6 for JacSketch is given in terms of the Lyapunov function:

$$\Psi^t = \|\mathbf{x}^t - \mathbf{x}^*\|^2 + \frac{\alpha}{2\mathcal{L}_2} \|\mathbf{J}^t - \nabla F(\mathbf{x}^*)\|_{\mathbf{W}^{-1}}^2,$$

where $\alpha$ is the step size used by the algorithm. Under appropriate choice of the step-size, Theorem 3.6 states that:

$$\mathbb{E}[\Psi^t] \leq \left(1 - \mu \min\left\{\frac{1}{4\mathcal{L}_1}, \frac{\kappa}{4\mathcal{L}_2 \rho/n^2 + \mu}\right\}\right)^t \cdot \Psi^0,$$

where $\kappa = \lambda_{\min}(\mathbb{E}[\Pi_{\mathbf{S}}])$ is the *stochastic condition number* analogous to the one defined for the Generalized Kaczmarz method, $n$ is the data size and parameters $\rho, \mathcal{L}_1, \mathcal{L}_2$ and $\mu$ are problem dependent constants defined in Theorem 3.6. Similarly as before, we can use our surrogate expressions for the expected residual projection to obtain a precise estimate for the stochastic condition number $\kappa$ under sub-gaussian sketching:

$$\kappa \simeq \bar{\kappa} := \frac{\lambda_{\min}(\mathbf{W})}{\lambda_{\min}(\mathbf{W}) + 1/\gamma} \quad \text{for} \quad \gamma > 0 \quad \text{s.t.} \quad \operatorname{tr} \mathbf{W}(\mathbf{W} + \tfrac{1}{\gamma}\mathbf{I})^{-1} = k.$$

**Corollary 5.** *Suppose $\mathbf{S}_t$ has size $k$ and i.i.d. mean-zero sub-gaussian entries. Let $r = \operatorname{tr} \mathbf{W}/\|\mathbf{W}\|$ be the stable rank of $\mathbf{W}^{\frac{1}{2}}$ and assume that $\rho = r/k$ is a constant larger than 1. Then,*

$$|\kappa - \bar{\kappa}| \leq O\left(\tfrac{1}{\sqrt{r}}\right).$$

### B.4   Omitted proofs

**Proof of Corollary 3**  Using Theorem 1, for $\bar{\mathbf{P}}_\perp$ as defined in (1), we have

$$(1 - \epsilon)\bar{\mathbf{P}}_\perp \preceq \mathbf{I} - \mathbb{E}[\mathbf{P}] = \mathbb{E}[\mathbf{P}_\perp] \preceq (1 + \epsilon)\bar{\mathbf{P}}_\perp, \quad \text{where} \quad \epsilon = O\left(\tfrac{1}{\sqrt{r}}\right).$$

In particular, this implies that $\|\bar{\mathbf{P}}_\perp^{-\frac{1}{2}}(\mathbb{E}[\mathbf{P}_\perp] - \bar{\mathbf{P}}_\perp)\bar{\mathbf{P}}_\perp^{-\frac{1}{2}}\| \leq \epsilon$. Moreover, in the proof of Theorem 2 we showed that $\frac{\rho-1}{\rho}\mathbf{I} \preceq \bar{\mathbf{P}}_\perp \preceq \mathbf{I}$, see (7), so it follows that:

$$\bar{\mathbf{P}}_\perp^{-1}(\mathbb{E}[\mathbf{P}_\perp] - \bar{\mathbf{P}}_\perp)^2 \bar{\mathbf{P}}_\perp^{-1} \preceq \frac{\rho}{\rho - 1}\left(\bar{\mathbf{P}}_\perp^{-\frac{1}{2}}(\mathbb{E}[\mathbf{P}_\perp] - \bar{\mathbf{P}}_\perp)\bar{\mathbf{P}}_\perp^{-\frac{1}{2}}\right)^2 \preceq \frac{\rho}{\rho - 1}\epsilon^2 \cdot \mathbf{I},$$

where note that $\frac{\rho}{\rho-1}\epsilon^2 = O(1/r)$, since $\rho$ is treated as a constant. Thus we conclude that:

$$\|\mathbb{E}[\Delta_{t+1}] - \bar{\Delta}_{t+1}\|^2 = \mathbb{E}[\Delta_t]^\top (\mathbb{E}[\mathbf{P}_\perp] - \bar{\mathbf{P}}_\perp)^2 \mathbb{E}[\Delta_t]$$
$$\leq O(1/r) \cdot \mathbb{E}[\Delta_t]^\top \bar{\mathbf{P}}_\perp^2 \mathbb{E}[\Delta_t] = O(1/r) \cdot \|\bar{\Delta}_{t+1}\|^2,$$

which completes the proof of (17). To show (18), it suffices to observe that

$$\lambda_{\min}(\mathbb{E}[\mathbf{P}]) = 1 - \lambda_{\max}(\mathbb{E}[\mathbf{P}_\perp]) \geq 1 - (1 + \epsilon)\lambda_{\max}(\bar{\mathbf{P}}_\perp) \geq \lambda_{\min}(\mathbf{I} - \bar{\mathbf{P}}_\perp) - \epsilon,$$

which completes the proof since $\mathbf{I} - \bar{\mathbf{P}}_\perp = \gamma\mathbf{A}^\top\mathbf{A}(\gamma\mathbf{A}^\top\mathbf{A} + \mathbf{I})^{-1}$. ∎

Corollaries 4 and 5 follow analogously from Theorem 1.

Figure 4: Theoretical predictions versus approximation error for the sketched Nyström with the RBF kernel, using Gaussian and Rademacher sketches (spectral decay shown at the bottom).

# C    Additional empirical results

We complement the results of Section 4 with empirical results on four additional libsvm datasets [CL11] (bringing the total number of benchmark datasets to eight), which further establish the accuracy of our surrogate expressions for the low-rank approximation error. Similary as in Figure 2, we use the sketched Nyström method [GM16] with the RBF kernel $k(\mathbf{a}_i, \mathbf{a}_j) = \exp(-\|\mathbf{a}_i - \mathbf{a}_j\|^2/(2\sigma^2))$, for several values of the parameter $\sigma$. The values of $\sigma$ were chosen so as to demonstrate the effectiveness of our theoretical predictions both when the stable rank is moderately large and when it is very small.

In Figure 4 we show the results for both Gaussian and Rademacher sketches. These results reinforce the conclusions we made in Section 4: our theoretical estimates are very accurate in all cases, for both sketching methods, and even when the stable rank is close to 1 (a regime that is not supported by the current theory).