[Reviews · NeurIPS 2020]

Review 1

Summary and Contributions: This paper proposes a semi-explicit expression for the expected value of a random projection matrix involved in sketching. More precisely, if S is the sketch matrix for A the data matrix, then in order to investigate the error of sketching one needs to control \Porth=I-P, where P is the orthogonal projection on the subspace spanned by the rows of SA. The main result of the paper is that \expec{\Porth}=(\gamma A^\top A+I)^{-1} up to error terms depending on the stable rank of A, and where \gamma is solution of some equation. Some applications of this results are given in the field of randomized iterative optimization, and the validity of the main result is verified by several numerical experiments.

Strengths: *after authors' response:* Thanks for the response, I will keep my score. -------------------------------------------------------------------------------------------------------------- The theoretical result is new, and has very interesting corollaries for randomized iterative optimization schemes (namely, trivializing the convergence proofs). Another application of the result is implicit regularization: the paper shows that Theorem 1 can be used to study the bias of the \ell_2 regularized solution for least squares. The exposition of the theoretical results is very neat, and quite understandable. Some care was also given in demonstrating numerically the peertinence of the main result in simple, understandable cases.

Weaknesses: In my opinion, the main caveat is the non expliciteness of \gamma. While this is seriously adressed by the paper in Section 4, I feel like I still do not understand well the behavior of k after reading the paper.

Correctness: As far as I can tell, the proof of the theoretical results are correct. One minor thing: line 87, the paper states that "\gamma increases at least linearly as a function of $k$." Even if the proof is not complicated, I think it deserves to be explained to help the reader get intuition on \gamma. Simple bounds, and maybe a graph, would also help.

Clarity: The paper is very well-written.

Relation to Prior Work: The prior work is appropriately cited. Maybe it would just be better to cite Hanson and Wright when Hanson-Wright inequality is used in Lemma 2 even though the version from Zajkowski (2018) is used.

Reproducibility: Yes

Additional Feedback: This is just a suggestion, but I think that Section 1.3 should be expanded. The current version of the paper goes a bit quickly there for readers that are not familiar with randomized iterative optimization, double descent, or both topics. It is especially hard to see from where the equation line 136 comes from. This could be achieved easily by removing one of the two proof sketches of Theorem 2 (page 5): one of them is enough. Another suggestion: since the closed-form expressions of \gamma are obtained in the n,\to +\infty limit, it looks possible to consider centered i.i.d. data and use Marcenko-Pastur to approximate the sum in (4), therefore obtaining another expression for \gamma in this case.


Review 2

Summary and Contributions: This paper is concerned with sketching of matrices, where the authors use subgaussian sketching matrices. The authors derive accurate predictions for the residual projection matrix. This allows them to make accurate prediction for the low-rank approximation error. Furthermore, the authors conduct simulations, which confirm the predictions of their theory. ----------------------------------------- I want to thank the authors for their rebuttal and for answering my questions. I will keep my score.

Strengths: This work characterizes the low-rank approximation error precisely with respect to the singular values of the sketched matrix. Since their guarantee depends precisely on the distribution of the existing singular values, this goes significantly beyond existing literature, where typically worst-case guarantees are discussed. This makes this paper an important contribution in the sketching literature.

Weaknesses: -Theorem 1 and Theorem 2 only hold with high probability according to the proofs. This is missing in the statement of the both theorems. The authors should add this to the statement. -In order to quantify the low-rank approximation error the authors use the quantity E||A-AP||^2. Since this is central to the paper, it would be great if the authors could add more explanation and motivation to it (and maybe mention other notions).

Correctness: The claims are sound and the arguments in the main part are correct.

Clarity: The paper is clearly written.

Relation to Prior Work: Prior work is clearly discussed.

Reproducibility: Yes

Additional Feedback: What is the high-level motivation of relying on the rank-one update formula in the proof (see Lemma 1)?


Review 3

Summary and Contributions: This paper offers spectral bounds when sketching matrices using certain types of random random projections. The results are novel, compared to literature, in that the authors give bounds in terms of the stable rank. As a special case, the bounds improve if the spectral gap is large enough.

Strengths: Most random projections are analyzed in the worst case. This is usually not particularly surprising, as there is no worst or best case in most regimes. This is also their feature, as they are data oblivious, meaning that any special structure is difficult to exploit. As such, identifying a setting where there are better bounds (e.g. spectral gap) is a very nice result.

Weaknesses: The related work could be done better. Matrix sketching is a very large field with plenty of work. While the selection of citations are ok, I wish the authors would have placed greater emphasis on previous results with regards to subspace embeddings, especially when placing their own results. This should not necessarily impact the paper's chances of getting accepted, but nevertheless I hope that the authors do that for the final version

Correctness: yes

Clarity: yes

Relation to Prior Work: absolutely inadequate.

Reproducibility: Yes

Additional Feedback:

[Author Response · NeurIPS 2020]

Two reviewers rated this paper very highly. The other reviewer was anomalously low and provided a review that lacked substance and was completely inappropriate. We thus ask the senior PC member to discount this review and make a decision based on the other two thoughtful reviews, or to find another reviewer to provide a more appropriate review. (We will, however, address to the extent possible the comments of this reviewer below.)

**To Reviewer #2** Thanks for the clear accept and the helpful suggestions. In the final version, we will clarify the discussion of $\gamma$ as a function of $k$ in Section 1.2, and we will also expand Section 1.3 (note that a detailed discussion of randomized iterative optimization is provided in Appendix B).

**To Reviewer #3** Thanks for the clear accept and the helpful suggestions. It is true that our proofs define a high probability event, however this is merely for the analysis. The final statements of Theorems 1 and 2 hold absolutely, rather than with high probability. In the final version, we will expand our discussion of low-rank approximation metrics.

**Addressing Reviewer #4** Below, we demonstrate that the review is inappropriate and should be discounted, because:

1. The reviewer completely misrepresents and fails to understand the stated aims of the paper.
2. The reviewer makes sweeping claims which are technically wrong and unsupported.

First of all, while we agree that the criticisms voiced by the reviewer apply to many NeurIPS papers, this paper is *not one of them*. We are well aware of the TCS work on low-rank approximation and sketching. If we were to cite all of those TCS papers, it would include hundreds of references (we choose to cite a few reviews, as is common when an area gets to a certain level of maturity). However, more importantly, unlike the TCS papers, in this work we are *not* interested in obtaining worst-case approximation or concentration bounds (see, e.g., lines 4 and 27). Yet, the reviewer appears to be confused about this and states that "the authors show that, given a matrix A and a sketch S, the difference of the singular values of $A^T A$ and $A^T S^T S A$ is concentrated around the expectation." This is *not at all* accurate in describing our results. Instead, our goal is to provide a precise characterization, which goes *beyond worst-case bounds*, for the expected residual projection matrix, $\mathbb{E}[I - (SA)^\dagger SA]$ (i.e., approximating an analytically intractable deterministic quantity with a simpler analytically tractable expression). Thus, in the context of low-rank approximation, our goal is not to improve on a TCS-style approximation objective (e.g., by showing a $1 + \epsilon$ error bound relative to the best rank $k$ subspace), but rather to express the error in a simple form as a function of the spectrum of the data matrix. Also, unlike standard worst-case analysis, our analysis does *not* rely on satisfying some notion of the *subspace embedding* property, which significantly differentiates our work from that cited by the reviewer. Note that a subspace embedding is neither sufficient nor necessary for many numerical implementations of sketching [Avron et al., 2010, Meng et al., 2014], or statistical results [Raskutti and Mahoney, 2016, Dobriban and Liu, 2019, Yang et al., 2020], as well as in the context of iterative optimization and implicit regularization (see Section 1.3), which are discussed in detail in the paper.

Finally, we point out the false and unsupported claims made by the reviewer when comparing our paper to prior work. This likely arises from the reviewer's confusion regarding the nature of our results. (Meanwhile, the other two reviewers describe the paper as "very well-written" and "clearly written".) Regarding Cohen et al. [2016], Reviewer #4 states that "Main result here is a generalization of theorem 1", and then later, regarding the submission, the reviewer claims that "the results presented here are, in one form or another, either known results from TCS literature, or easy corollaries". The latter statement is incredibly broad, completely unsupported and simply false, so we focus on the former. Regarding the former claim, Cohen et al. do provide a low-rank approximation guarantee for sub-Gaussian sketches. However, this result differs from ours in several respects. First of all, instead of analyzing $\tilde{A} = A(SA)^\dagger SA$ directly as a low-rank approximation with a sketch of size $k$ (as we do), they use a larger sketch (e.g., of size $Ck/\epsilon^2$, where $C > 1$ and $\epsilon < 1$) and consider the matrix $\tilde{A}_k$, defined as the best rank $k$ approximation of $\tilde{A}$. This distinction is crucial for their analysis, which relies on showing that a sketch of size sufficiently larger than $k$ ensures a rank $k$ subspace embedding condition. This condition is not known to hold (at least in the worst case) if the sketch is of size $k$. Our novel analysis completely avoids subspace embeddings (which, as the reviewer points out, are central to TCS-style analysis). This is why we can still provide upper/lower bounds for the low-rank approximation error in this important case. Also, the form of our bounds is completely different than that of Cohen et al., in that we compare the error with a certain implicit function of the singular values of $A$, which is different from the error of the best rank $k$ approximation (used by Cohen et al.), and so the role of $\epsilon$ in our paper is different than in theirs. All in all, different methods are being analyzed, different types of bounds are obtained, and completely different analysis is used. Thus, by all indications, the reviewer is wrong, and our result is *not* a corollary or a special case of the results of Cohen et al.

# References

H. Avron, P. Maymounkov, and S. Toledo. Blendenpik: Supercharging lapack's least-squares solver. *SIAM Journal on Scientific Computing*, 32(3):1217–1236, 2010.

M. B. Cohen, J. Nelson, and D. P. Woodruff. Optimal approximate matrix product in terms of stable rank. In *43rd International Colloquium on Automata, Languages, and Programming, ICALP 2016, July 11-15, 2016, Rome, Italy*, pages 11:1–11:14, 2016.

E. Dobriban and S. Liu. Asymptotics for sketching in least squares regression. In *Advances in Neural Information Processing Systems*, pages 3675–3685, 2019.

X. Meng, M. A. Saunders, and M. W. Mahoney. LSRN: A parallel iterative solver for strongly over- or under-determined systems. *SIAM Journal on Scientific Computing*, 36(2):C95–C118, 2014.

G. Raskutti and M. W. Mahoney. A statistical perspective on randomized sketching for ordinary least-squares. *Journal of Machine Learning Research*, 17(214):1–31, 2016.

F. Yang, S. Liu, E. Dobriban, and D. P. Woodruff. How to reduce dimension with pca and random projections? *arXiv preprint arXiv:2005.00511*, 2020.


[Meta-Review · NeurIPS 2020]

Three knowledgeable referees recommend accept, considering this an important contribution that goes significantly beyond the literature in sketching. I also recommend accept. However, please consider revising your paper to address R4's remarks on related literature.